# Stabilizing LTI Systems under Partial Observability: Sample Complexity and Fundamental Limits

**Ziyi Zhang, Yorie Nakahira, Guannan Qu**
Department of Electrical and Computer Engineering
Carnegie Mellon University
Pittsburgh, PA 15213
`ziyizhan,yorie,gqu@andrew.cmu.edu`

## Abstract

We study the problem of stabilizing an unknown partially observable linear time-invariant (LTI) system. For fully observable systems, the state-of-the-art approaches leverage an unstable/stable subspace decomposition to achieve sample complexity that depends only on the number of unstable modes, independent of the dimension of the system state. However, it remains open whether such sample complexity can be achieved for partially observable systems because such systems do not admit a uniquely identifiable unstable subspace. In this paper, we propose LTS-P, a novel technique that leverages compressed singular value decomposition (SVD) on the "lifted" Hankel matrix to estimate the unstable subsystem up to an unknown transformation. Then, we design a stabilizing controller that integrates a robust stabilizing controller for the unstable mode and a small-gain-type assumption on the stable subspace. We show that LTS-P achieves state-of-the-art, dimension-free sample complexity that scales only with the number of unstable modes. This substantially reduces data requirements for stabilizing high-dimensional systems, particularly those dominated by stable dynamics.

## 1 Introduction

Learning-based control of unknown dynamical systems is of critical importance for many autonomous control systems [3, 30, 6, 24, 12]. Despite recent advances, many existing methods make strong assumptions such as open-loop stability, availability of an initial stabilizing controller, and fully observable systems [15, 41]. However, these assumptions may not hold in practice. Motivated by this gap, this paper studies the problem of stabilizing an unknown, partially-observable, unstable system without access to an initial stabilizing controller. In particular, we consider the following linear time-invariant (LTI) system:

$$
\begin{aligned}
x_{t+1} &= Ax_t + Bu_t \\
y_t &= Cx_t + Du_t + v_t,
\end{aligned}
\tag{1}
$$

where $x_t \in \mathbb{R}^n$ and $u_t \in \mathbb{R}^{d_u}, y_t \in \mathbb{R}^{d_y}$ are the state, control input, and observed output at time step $t \in \{0, \ldots, T-1\}$, respectively. The system also has additive observation noise $v_t \sim \mathcal{N}(0, \sigma_v^2 I)$. While there are works studying system identification for partially observable LTI systems [15, 43, 41, 34, 51], they do not address the problem of stabilization, and many assume the system is open-loop stable [15, 41, 34]. Other adaptive control approaches can address the learn-to-stabilize problem for fully observable systems based on Lyapunov methods [35, 37], but there are few systematic approaches to construct a Lyapunov function in a way that optimizes sample complexities or transient performance during learning.

In the special case of *fully-observable* LTI system ($C = I$), Chen and Hazan [9] reveals that the transient performance during the learn-to-stabilize process suffers from exponential blow-up, i.e. the

39th Conference on Neural Information Processing Systems (NeurIPS 2025).

system state can blow up exponentially in the state dimension. This phenomenon arises because stabilization requires accurate identification of the full system dynamics, which in turn necessitates at least $n$ samples along a single trajectory. During this identification phase, the system can remain unstable and grow exponentially. To relieve this problem, Hu et al. [19] proposed a framework that separates the unstable component and focuses on stabilizing these subsystems. This reduces the sample complexity to only grow with the number of unstable eigenvalues, rather than the full state dimension $n$. This result was later extended to noisy setting in Zhang et al. [50]. To date, this dependence on the number of unstable eigenvalues remains the best sample complexity for the learn-to-stabilize problem in the fully observable case.

In contrast, we address a more challenging setting of partially observable systems, and answer the following research question: *Is it possible to stabilize a partially observable LTI system by only identifying an unstable component (to be properly defined) with a sample complexity independent of the overall state dimension*? This question introduces two key technical challenges. First, partially observable systems typically require a dynamic controller, which renders the stable feedback controllers in the existing approach inapplicable [19]. Second, the construction in "unstable subspace" in Hu et al. [19] is not uniquely defined in the partially observable case. This is because the state and the $A, B, C$ matrices are not uniquely identifiable from the learner's perspective, as any similarity transformation of the matrices can yield the same input-output behavior.

**Contribution.** In this paper, we answer the above question in the affirmative. Firstly, we propose a novel definition of "unstable component" to be a low-rank version of the transfer function of the original system that only retains the unstable eigenvalues. Based on this unstable component, we propose LTS-P, which leverages compressed singular value decomposition (SVD) on a "lifted" Hankel matrix to estimate the unstable component. Then, we design a robust stabilizing controller for the unstable component, and show that it stabilizes the full system under a small-gain-type assumption on the $H_\infty$ norm of the stable component. Importantly, our theoretical analysis shows that the sample complexity of the proposed approach only scales with the dimension of unstable component, i.e. the number of unstable eigenvalues, as opposed to the dimension of the full state space. We also conduct simulations to validate the effectiveness of LTS-P, showcasing its ability to efficiently stabilize partially observable LTI systems with reduced samples (Figures 1 and 2).

Moreover, the technical innovations underlying our approach are of independent interest. We show that by using compressed singular value decomposition (SVD) on a properly defined lifted Hankel matrix, we can estimate the unstable component of the system. While this is conceptually related to classical model reduction techniques [13, Chapter 4.6], our setting departs significantly: unlike prior work for stable dynamics and bounded Hankel matrices [13], we address systems with unstable modes, where the associated Hankel matrices may grow unboundedly over time. This distinction renders standard identification and reduction techniques inapplicable [15, 41, 34]. Interestingly, the $H_\infty$ norm condition on the stable component, derived from the small gain theorem, is a necessary and sufficient condition for stabilization. This characterization reveals the exact subspace of the system that must be estimated for stabilization to be possible. As a result, our analysis not only supports the optimality of LTS-P in terms of sample complexity but also informs the fundamental limit of stabilizability.

**Related Work.** Our work is closely related to learn-to-stabilize on multiple trajectories and learn-to-control with known stabilizing controllers. In addition, we will also briefly cover existing literature in learn-to-stabilize on a single trajectory, adaptive control, and system identification.

*Learn-to-stabilize on multiple trajectories.* There are also works that do not assume open-loop stability and learn the full system dynamics before designing a stabilizing controller, typically incurring a sample complexity of $\widetilde{\Theta}(\text{poly}(n))$ [12, 45, 51]. Recently, a model-free approach via the policy gradient method offers a novel perspective with the same sample complexity [36]. While these techniques are developed for fully observable systems, the proposed algorithm, LTS-P, tackles a significantly more challenging setting of partially observable, unstable systems. As detailed above, this setting introduces fundamental technical challenges that require novel algorithmic and analytical tools beyond those used in fully observable systems. Moreover, compared with those works, the sample complexity of the proposed algorithm does not depend on $n$ and only scales with the number of unstable modes.

*Learn to control with known stabilizing controller.* Extensive research has been conducted on stabilizing LTI systems under stochastic noise [5, 22, 25, 30]. One branch of research uses the

model-free approach to learn the optimal controller [16, 21, 30, 46, 49]. Those algorithms typically require a known stabilizing controller as an initialization policy for the learning process. Another line of research utilizes model-based approaches, which require an initial stabilizing controller to learn the system dynamics before designing the controllers [10, 32, 38, 52]. On the other hand, we focus on learn-to-stabilize. Our method can be used as the initial policy in these methods to remove their requirement for initial stabilizing controllers.

*Learn-to-stabilize on a single trajectory.* Learning to stabilize a linear system in an infinite time horizon is a classic problem in control [7, 26, 27]. Early works achieved the regret bounds of $2^{O(n)}O(\sqrt{T})$, which rely on assumptions of observability and strictly stable transition matrices [1, 20]. Subsequent work improved the regret to $2^{\tilde{O}(n)} + \tilde{O}(\text{poly}(n)\sqrt{T})$ [9, 28]. Recently, Hu et al. [19] proposed an algorithm that requires $\tilde{O}(k)$ samples, where $k$ is the number of unstable modes. While these techniques are developed for fully observable systems using a single trajectory, we consider a different problem of learning to stabilize a partially observable system using multiple trajectories.

*Adaptive control.* Adaptive control is a well-established methodology for controlling systems with uncertain or time-varying parameters [8, 35, 37]. Existing literature has established techniques for stabilizing unknown systems with asymptotic stability [9, 14, 29, 44, 45]. Other works use past trajectory to estimate the system dynamics and then design the controller [4, 11, 31]. While these works often assume stability, fully observable systems and are based on Lyapunov methods, we focus on partially observable, unstable systems and build our algorithm based on compressed SVD on the lifted Hankel matrix.

*System identification.* Existing literature has developed a variety of techniques to estimate system parameters [33, 40, 43, 42, 47]. Hankel matrix are also used in techniques such as Eigensystem Realization Algorithm (ERA), subspace identification, etc [23, 39]. Our work utilizes a similar approach to partially determine the system parameters before constructing the stabilizing controller. While these works focus on identifying the system dynamics, we close the loop and establish state-of-the-art and optimal sample complexity guarantees for the stabilization problem.

## 2 Problem Statement

**Notations.** We use $\|\cdot\|$ to denote the $L^2$-norm for vectors and the spectral norm for matrices. We use $M^*$ to represent the conjugate transpose of $M$. We use $\sigma_{\min}(\cdot)$ and $\sigma_{\max}(\cdot)$ to denote the smallest and largest singular value of a matrix, and $\kappa(\cdot)$ to denote the condition number of a matrix. We use the standard big $O(\cdot)$, $\Omega(\cdot)$, $\Theta(\cdot)$ notation to highlight dependence on a certain parameter, hiding all other parameters. We use $f \lesssim g$, $f \gtrsim g$, $f \asymp g$ to mean $f = O(g)$, $f = \Omega(g)$, $f = \Theta(g)$ respectively while *only hiding numeric constants*. We provide an indexing of notations at Appendix I.

For simplicity, we primarily deal with the system where $D = 0$. For the case where $D \neq 0$, we can easily estimate $D$ in the process and subtract $Du_t$ to obtain a new observation measure not involving control input. We briefly introduce the method for estimating $D$ and how to apply the proposed algorithm in the case when $D \neq 0$ in Appendix F.

**Learn-to-stabilize.** As the unknown system as defined in (1) can be unstable, the goal of the learn-to-stabilize problem is to return a controller that stabilizes the system using data collected from interacting with the system on $M$ rollouts. More specifically, in each rollout, the learner can determine $u_t$ and observe $y_t$ for a rollout of length $T$ starting from $x_0$, which we assume $x_0 = 0$ for simplicity of proof.

**Goal.** The sample complexity of stabilization is the number of samples, $MT$, needed for the learner to return a stabilizing controller. Standard system identification and certainty equivalence controller design need at least $\Theta(n)$ samples for stabilization, as $\Theta(n)$ is the number of samples needed to learn the full dynamical system. In this paper, our goal is to study whether it is possible to stabilize the system with sample complexity independent from $n$.

### 2.1 Background on $H_\infty$ control

In this section, we briefly introduce the background of $H$-infinity control. First, we define the open loop transfer function of system (1) from $u_t$ to $y_t$ to be

$$F^{\text{full}}(z) = C(zI - A)^{-1}B, \tag{2}$$

which reflects the cumulative output of the system in the infinite time horizon. Next, we introduce the $\mathcal{H}_\infty$ space on transfer functions in the $z$-domain.

**Definition 2.1** ($\mathcal{H}_\infty$-space)**.** Let $\mathcal{H}_\infty$ denote the Banach space of matrix-valued functions that are analytic and bounded outside of the unit sphere. Let $\mathcal{RH}_\infty$ denote the real and rational subspace of $\mathcal{H}_\infty$. The $\mathcal{H}_\infty$-norm is defined as

$$\|f\|_{\mathcal{H}_\infty} := \sup_{z \in \mathbb{C}, |z| \geq 1} \sigma_{\max}(f(z)) = \sup_{z \in \mathbb{C}, |z|=1} \sigma_{\max}(f(z)),$$

where the second equality is a simple application of the Maximum modulus principle. We also denote $\mathbb{C}_{\geq 1} = \{z \in \mathbb{C} : |z| \geq 1\}$ be the complement of the unit disk in the complex domain. For any transfer function $G$, we say it is *internally stable* if $G \in \mathcal{RH}_\infty$.

The $H_\infty$ norm of a transfer function is crucial in robust control, as it represents the amount of modeling error the system can tolerate without losing stability, due to the small gain theorem [53]. Abundant research has been done in $H_\infty$ control design to minimize the $H_\infty$ norm of transfer functions [53]. In this work, $H_\infty$ control play an important role as we treat the stable component (to be defined later) of the system as a modeling error and show that the control we design can stabilize despite the modeling error.

## 3 Algorithm Idea

In this paper, we assume the matrix $A$ does not have marginally stable eigenvalues, and the eigenvalues are ranked in decreasing order of magnitude, i.e. $|\lambda_1| \geq |\lambda_2| \geq \cdots \geq |\lambda_k| > 1 > |\lambda_{k+1}| \geq \cdots \geq |\lambda_n|$. The high-level idea of the paper is to first decompose the system dynamics into the unstable and stable components (Section 3.1), estimate the unstable component via a low-rank approximation of the lifted Hankel matrix (Section 3.2), and design a robust stabilizing for the unstable component that stabilizes the whole system (Section 3.3).

### 3.1 Decomposition of Dynamics

Given the eigenvalues, we have the following decomposition for the system dynamics matrix:

$$A = \underbrace{[Q_1 Q_2]}_{:=Q} \begin{bmatrix} N_1 & 0 \\ 0 & N_2 \end{bmatrix} \underbrace{\begin{bmatrix} R_1 \\ R_2 \end{bmatrix}}_{:=R}, \tag{3}$$

where $R = Q^{-1}$, and the columns of $Q_1 \in \mathbb{R}^{n \times k}$ are an orthonormal basis for the invariant subspace of the unstable eigenvalues $\lambda_1, \ldots, \lambda_k$, with $N_1$ inheriting eigenvalues $\lambda_1, \ldots, \lambda_k$ from $A$. Similarly, columns of $Q_2 \in \mathbb{R}^{n \times (n-k)}$ form an orthonormal basis for the invariant subspace of the unstable eigenvalues $\lambda_{k+1}, \ldots, \lambda_n$ and $N_2$ inherit all the stable eigenvalues $\lambda_{k+1}, \ldots, \lambda_n$.

Given the decomposition of the matrix $A$, our key idea is to only estimate the unstable component of the dynamics, which we define below. Consider the transfer function of the original system:

$$F^{\text{full}}(z) = C(zI - A)^{-1}B = C \left(zI - [Q_1 Q_2] \begin{bmatrix} N_1 & 0 \\ 0 & N_2 \end{bmatrix} \begin{bmatrix} R_1 \\ R_2 \end{bmatrix} \right)^{-1} B$$

$$= C \left([Q_1 Q_2] \begin{bmatrix} zI - N_1 & 0 \\ 0 & zI - N_2 \end{bmatrix} \begin{bmatrix} R_1 \\ R_2 \end{bmatrix} \right)^{-1} B$$

$$= \underbrace{CQ_1(zI - N_1)^{-1}R_1 B}_{:=F(z)} + \underbrace{CQ_2(zI - N_2)^{-1}R_2 B}_{:=\Delta(z)} = F(z) + \Delta(z). \tag{4}$$

Therefore, the original system is an additive decomposition into the unstable transfer function $F(z)$, which we refer to as the unstable component, and the stable transfer function $\Delta(z)$, which we refer to as the stable component.

### 3.2 Approximate low-rank factorization of the lifted Hankel matrix

In this section, we define a "lifted" Hankel matrix, show it admits a rank $k$ approximation, based on which the unstable component can be estimated.

If each rollout has length $T$, we can decompose $T := m+p+q+2$ and estimate the following "lifted" Hankel matrix where the $(i,j)$-th block is $[H]_{ij} = CA^{m+i+j-2}B$ where $i = 1, \ldots, p$, $j = 1, \ldots, q$. In other words,

$$
H = \begin{bmatrix}
CA^m B & CA^{m+1}B & \cdots & CA^{m+q-1}B \\
CA^{m+1}B & CA^{m+2}B & \cdots & CA^{m+q}B \\
\cdots & \cdots & \cdots & \cdots \\
CA^{m+p-1}B & CA^{m+p}B & \cdots & CA^{m+p+q-2}B
\end{bmatrix}, \tag{5}
$$

for some $m, p, q$ that we will select later. We call this Hankel matrix "lifted" as it starts with $CA^m B$. This "lifting" is essential to our approach, as raising $A$ to the power of $m$ can separate the stable and unstable components and facilitate better estimation of the unstable component, which will become clear later on.

Define

$$
\mathcal{O} = \begin{bmatrix}
CA^{m/2} \\
CA^{m/2+1} \\
\cdots \\
CA^{m/2+p-1}
\end{bmatrix}, \qquad \mathcal{C} = [A^{m/2}B, A^{m/2+1}B, \cdots, A^{m/2+q-1}B]. \tag{6}
$$

Then we have the factorization $H = \mathcal{O}\mathcal{C}$, indicating that $H$ is of rank at most $n$.

**Rank $k$ approximation.** We now show that $H$ has a rank $k$ approximation corresponding to the unstable component. Given the decomposition of $A$ in (3), we can write each block of the lifted Hankel matrix as

$$
CA^\ell B = C[Q_1 Q_2] \begin{bmatrix} N_1^\ell & 0 \\ 0 & N_2^\ell \end{bmatrix} \begin{bmatrix} R_1 \\ R_2 \end{bmatrix} B = [CQ_1 \quad CQ_2] \begin{bmatrix} N_1^\ell & 0 \\ 0 & N_2^\ell \end{bmatrix} \begin{bmatrix} R_1 B \\ R_2 B \end{bmatrix}.
$$

If $\ell$ is resonably large, using the fact that $N_1^\ell \gg N_2^\ell \approx 0$, we can have $CA^\ell B \approx CQ_1 N_1^\ell R_1 B$. Therefore, we know that when $m$ is reasonably large, we have $H$ can be approximately factorized as $H \approx \tilde{\mathcal{O}}\tilde{\mathcal{C}}$ where:

$$
\tilde{\mathcal{O}} = \begin{bmatrix}
CQ_1 N_1^{m/2} \\
CQ_1 N_1^{m/2+1} \\
\cdots \\
CQ_1 N_1^{m/2+p-1}
\end{bmatrix}, \qquad \tilde{\mathcal{C}} = [N_1^{m/2}R_1 B, \ldots, N_1^{m/2+q-1}R_1 B]. \tag{7}
$$

As $\tilde{\mathcal{O}}$ has $k$ columns, $\tilde{\mathcal{O}}\tilde{\mathcal{C}}$ has (at most) rank-$k$. We also use the notation

$$
\tilde{H} = \tilde{\mathcal{O}}\tilde{\mathcal{C}}, \tag{8}
$$

to denote this rank-$k$ approximation of Hankel. As from $H$ to $\tilde{H}$, the only thing that are omitted are of order $O(N_2^m)$, it is reasonable to expect that $\|H - \tilde{H}\| \le O(\lambda_{k+1}^m)$, i.e. this rank $k$ approximation has exponentially decaying error in $m$, as shown in Lemma C.3.

**Estimating unstable component of dyamics.** In the actual algorithm (to be introduced in Section 4), $\tilde{H}$ is to be estimated and therefore not known perfectly. However, to illustrate the methodology of the proposed method, for this subsection, we consider $\tilde{H}$ to be known perfectly and show that the unstable component $F(z)$ can be recovered perfectly.

Suppose we have the following factorization of $\tilde{H}$ for some $\bar{\mathcal{O}} \in \mathbb{R}^{d_y \times k}, \bar{\mathcal{C}} \in \mathbb{R}^{k \times d_u}$, (which has infinite possible solutions), $\tilde{H} = \bar{\mathcal{O}}\bar{\mathcal{C}}$. We show in Lemma D.1 there exists an invertible $S$, such that $\bar{\mathcal{O}} = \tilde{\mathcal{O}}S$, $\bar{\mathcal{C}} = S^{-1}\tilde{\mathcal{C}}$. Therefore, from the construction of $\tilde{\mathcal{O}}$ and $\tilde{\mathcal{C}}$ in (7), we see that $\bar{\mathcal{O}}_1 = CQ_1 N_1^{m/2}S$, where $\bar{\mathcal{O}}_1$ represent the first block submatrix of $\bar{\mathcal{O}}$.[1] Solving for $\bar{N}_1$ for the equation $\bar{\mathcal{O}}_{2:p} = \bar{\mathcal{O}}_{1:p-1}\bar{N}_1$, we can get $\bar{N}_1 = S^{-1}N_1 S$. After which we can get $\bar{\mathcal{C}} = \bar{\mathcal{O}}_1 \bar{N}_1^{-m/2} = CQ_1 S$, and $\bar{B} = \bar{N}_1^{-m/2}\bar{\mathcal{C}}_1 = S^{-1}R_1 B$. In summary, we get the following realization of the system:

$$
z_{t+1} = S^{-1}N_1 S z_t + S^{-1}R_1 B u_t, \tag{9a}
$$

$$
y_t = CQ_1 S z_t. \tag{9b}
$$

whose transfer function is exactly $F(z)$, the unstable component of the original system.

---

[1] For row block or column block matrices $M$, we use $M_i$ to denote its $i$'th block, and $M_{i:j}$ to denote the submatrix formed by the $i, i+1, \ldots, j$'th block.

---
**Algorithm 1** LTS-P: learning the Hankel Matrix
---
1: **for** $i = 1 : M$ **do**
2:      Generate and apply $T$ Gaussian control inputs $[u_0^{(i)}, \ldots, u_{T-1}^{(i)}] \overset{\text{i.i.d}}{\sim} \mathcal{N}(0, \sigma_u^2)$. Collect $y^{(i)} =$
     $[y_0^{(i)}, \ldots, y_{T-1}^{(i)}]$.
3: **end for**
4: Compute $\hat{\Phi}$ with (17).
5: Recover $\hat{H}$ with (18).
6: Compute $\hat{\tilde{H}}$ from $\hat{H}$ with (19).
7: Compute $\hat{\mathcal{O}}$ and $\hat{\mathcal{C}}$ with (20).
8: Recover $\hat{N}_1, \hat{N}_{1,O}^{\frac{m}{2}}, \hat{N}_{1,C}^{\frac{m}{2}}$ with (21), (22), (23),respectively.
9: Compute $\hat{C}, \hat{B}$ with (24).
10: Design $H_\infty$ controller with $(\hat{N}_1, \hat{C}, \hat{B})$.
---

## 3.3 Robust stabilizing controller design

After the unstable component $F(z)$ is estimated, we can treat the unobserved part of the system as a disturbance $\Delta(z)$ and then synthesize a robust stabilizing controller. Suppose we design a controller $u(z) = K(z)y(z)$ for $F(z)$, and denote its sensitivity function as:

$$F_K(z) = (I - F(z)K(z))^{-1}.$$

Now let's look at what if we applied the same $K(z)$ to the full system $F^{\text{full}}(z)$ in (4). In this case, the closed loop system is stable if and only if the transfer function $F_K^{\text{full}}(z) = (I - F^{\text{full}}(z)K(z))^{-1}$ is analytic for all $|z| \geq 1$, see e.g. Chapter 5 of [53]. Note that,

$$F_K^{\text{full}}(z) = (I - F^{\text{full}}(z)K(z))^{-1} = (I - F(z)K(z) - \Delta(z)K(z))^{-1}$$
$$= (F_K(z)^{-1} - \Delta(z)K(z))^{-1} = F_K(z)(I - \Delta(z)K(z)F_K(z))^{-1}. \tag{10}$$

The Small Gain Theorem shows a necessary and sufficient condition for the system to have a stabilizing controller:

**Lemma 3.1** (Theorem 8.1 of [53]). *Given $\gamma > 0$, the closed loop transfer function defined in* (10) *is internally stable for any $\|\Delta(z)\|_{H_\infty} \leq \gamma$ if and only if $\|K(z)F_K(z)\|_{H_\infty} < \frac{1}{\gamma}$.*

Since $\Delta(z)$ is stable and therefore has bounded $H_\infty$ norm, it suffices to find a controller such that

$$\|K(z)F_K(z)\|_{H_\infty} < \frac{1}{\|\Delta(z)\|_{H_\infty}}, \tag{11}$$

in order to stabilize the original full system.

## 4 Algorithm

Based on the ideas developed in Section 3, we now design an algorithm that learns to stabilize from data. The pseudocode is provided in Algorithm 1.

**Step 1: Approximate Low-Rank Lifted Hankel Estimation.** Section 3 shows that if $\tilde{H}$ defined in (8) is known, then we can design a controller satisfying (11) to stabilize the system. In this section, we discuss a method to estimate $\tilde{H}$ with singular value decomposition (SVD) of the lifted Hankel metrix.

*Data collection and notation.* Consider we sample $M$ trajectories, each with length $T$. To simplify notation, for each trajectory $i \in \{1, \ldots, M\}$, we organize input and output data as

$$y^{(i)} = \begin{bmatrix} y_0^{(i)} & y_1^{(i)} & \ldots & y_{T-1}^{(i)} \end{bmatrix}, \qquad u^{(i)} = \begin{bmatrix} u_0^{(i)} & u_1^{(i)} & \ldots & u_{T-1}^{(i)} \end{bmatrix}, \tag{12}$$

where each $u_j^{(i)} \sim \mathcal{N}(0, \sigma_u^2)$ are independently selected. We also define the a new matrix for the observation noise as

$$v^{(i)} = \begin{bmatrix} v_0^{(i)} & v_1^{(i)} & \ldots & v_{T-1}^{(i)} \end{bmatrix}. \tag{13}$$

Substituting the above into (1), we obtain

$$y_t^{(i)} = C x_t^{(i)} + v_t^{(i)} = \sum_{j=0}^{T} C A^j \left( B u_{t-j-1}^{(i)} + w_{t-j-1}^{(i)} \right) + v_t^{(i)}. \tag{14}$$

We further define an upper-triangular Toeplitz matrix and a matrix of observed system dynamics:

$$U^{(i)} = \begin{bmatrix} u_0^{(i)} & u_1^{(i)} & u_2^{(i)} & \dots & u_{T-1}^{(i)} \\ 0 & u_0^{(i)} & u_1^{(i)} & \dots & u_{T-2}^{(i)} \\ \vdots & \vdots & \vdots & \ddots & \vdots \\ 0 & 0 & 0 & \dots & u_0^{(i)} \end{bmatrix}, \qquad \Phi = \begin{bmatrix} 0 & CB & CAB & \dots & CA^{T-2}B \end{bmatrix}. \tag{15}$$

Note that the lifted Hankel matrix $H$ in (5) can be recovered from $\Phi$ above as they contain the same block submatrices. The measurement data (14) in each rollout can be written as

$$y^{(i)} = \Phi U^{(i)} + v^{(i)}. \tag{16}$$

Therefore, we can estimate $\Phi$ by the following ordinary least square problem:

$$\hat{\Phi} := \arg \min_{X \in \mathbb{R}^{d_y \times (T * d_u)}} \sum_{i=1}^{M} \left\| y^{(i)} - X U^{(i)} \right\|_F^2. \tag{17}$$

We then estimate the lifted Hankel matrix as follows:

$$\hat{H} := \begin{bmatrix} \hat{\Phi}_{2+m} & \hat{\Phi}_{3+m} & \dots & \hat{\Phi}_{2+p+m} \\ \hat{\Phi}_{3+m} & \hat{\Phi}_{4+m} & \dots & \hat{\Phi}_{3+p+m} \\ \vdots & \vdots & \vdots & \vdots \\ \hat{\Phi}_{2+q+m} & \hat{\Phi}_{3+q+m} & \dots & \hat{\Phi}_{2+p+q+m} \end{bmatrix}. \tag{18}$$

Let $\hat{H} = \hat{U}_H \hat{\Sigma}_H \hat{V}_H^*$ denote the singular value decomposition of $\hat{H}$, and we define the $k$-th order estimation of $\hat{H}$:

$$\hat{\hat{H}} := \hat{U} \hat{\Sigma} \hat{V}^*, \tag{19}$$

where $\hat{U}$ ($\hat{V}$) is the matrix of the top $k$ left (right) singular vector in $\hat{U}_H$ ($\hat{V}_H$), and $\hat{\Sigma}$ is the matrix of the top $k$ singular values in $\hat{\Sigma}_H$.

**Step 2: Esitmating unstable transfer function F.** With the SVD $\hat{\hat{H}} = \hat{U} \hat{\Sigma} \hat{V}^*$, we further do the following factorization:

$$\hat{\mathcal{O}} = \hat{U} \hat{\Sigma}^{1/2}, \qquad \hat{\mathcal{C}} = \hat{\Sigma}^{1/2} \hat{V}^*. \tag{20}$$

With the above, we can estimate $\bar{N}_1, \bar{C}, \bar{B}$ similar to the procedure introduced in Section 3.2:

$$\hat{N}_1 := (\hat{\mathcal{O}}_{1:p-1}^* \hat{\mathcal{O}}_{1:p-1})^{-1} \hat{\mathcal{O}}_{1:p-1}^* \hat{\mathcal{O}}_{2:p}. \tag{21}$$

To reduce the error of estimating $CQ_1$ and $R_1 B$ and avoid compounding error caused by raising $\hat{N}_1$ to the $m/2$'th power, we also directly estimate $\bar{N}_1^{\frac{m}{2}}$ from both $\hat{\mathcal{O}}$ and $\hat{\mathcal{C}}$.

$$\widehat{N_{1,O}^{\frac{m}{2}}} := (\hat{\mathcal{O}}_{1:p-\frac{m}{2}}^* \hat{\mathcal{O}}_{1:p-\frac{m}{2}})^{-1} \hat{\mathcal{O}}_{1:p-\frac{m}{2}}^* \hat{\mathcal{O}}_{\frac{m}{2}+1:p}, \tag{22}$$

$$\widehat{N_{1,C}^{\frac{m}{2}}} := \hat{\mathcal{C}}_{\frac{m}{2}+1:p} \hat{\mathcal{C}}_{1:p-\frac{m}{2}}^* (\hat{\mathcal{C}}_{1:p-\frac{m}{2}} \hat{\mathcal{C}}_{1:p-\frac{m}{2}}^*)^{-1}. \tag{23}$$

In practice, the estimation obtained from (22) and (23) are very similar, and we can use either one to estimate $\bar{N}_1^{\frac{m}{2}}$. Lastly, we estimate $\bar{C}$ and $\bar{B}$ as follows:

$$\hat{C} := \hat{\mathcal{O}}_1 \widehat{N_{1,O}^{-\frac{m}{2}}}, \qquad \hat{B} := \widehat{N_{1,C}^{-\frac{m}{2}}} \hat{\mathcal{C}}_1, \tag{24}$$

where for simplicity, we use $\widehat{N_{1,O}^{-\frac{m}{2}}}, \widehat{N_{1,C}^{-\frac{m}{2}}}$ to denote the invserse of $\widehat{N_{1,O}^{\frac{m}{2}}}, \widehat{N_{1,C}^{\frac{m}{2}}}$. We will provide accuracy bounds for those estimations in Appendix B. With the above estimations, we are ready to design a controller with the following transfer function,

$$\hat{F}(z) = \hat{C}(zI - \hat{N}_1)^{-1} \hat{B}. \tag{25}$$

**Step 3: Designing robust controller.** After estimating the system dynamics and obtain the estimated transfer function in (25) as discussed in Section 3.3, we can design a stabilizing controller by existing $H_\infty$ control methods to minimize $\|K(z)F_K(z)\|_{H_\infty}$. The details on designing the robust controller can be found in robust control documentations, e.g. Chapter 7 of Dullerud and Paganini [13].

**What if $k$ is not known a priori?** The proposed method requires knowledge of $k$, the number of unstable eigenvalues. If $k$ is not known, we show in Lemma B.1 and Lemma C.3 that the first $k$ singular values of $\hat{H}$ and $H$ increase exponentially with $m$, and the remaining singular values decrease exponentially with $m$. Therefore, with a reasonably sized $m$ and if $p, q$ are chosen to be larger than $k$, there will be a large spectral gap among the singular values of $\hat{H}$. The learner can use the location of the spectral gap to determine the value of $k$.

## 5 Main Results

In this section, we provide the sample complexity needed for the proposed algorithm to return a stabilizing controller. We first introduce a standard assumption on controllabilty and observability.

**Assumption 5.1.** The LTI system $(A, B)$ is controllable, and $(A, C)$ is observable.

We also need an assumption on the existence of a controller that meets the small gain theorem's criterion (11).

**Assumption 5.2.** There exists a controller $K(z)$ that stabilizes plant $F(z)$, such that for some fixed $\epsilon_* > 0$, $\|K(z)F_K(z)\|_{H_\infty} < \frac{1}{\|\Delta(z)\|_{H_\infty}+3\epsilon_*}$.

To state our main result, we introduce some system theoretic quantity.

**Controllability/observability Gramian for unstable subsystem.** For the unstable subsystem $N_1, R_1B, CQ_1$, its $\alpha$-observability Gramian is $\mathcal{G}_{\mathrm{ob}} = \left[(CQ_1)^* \quad (CQ_1N_1)^* \quad \ldots \quad (CQ_1N_1^{\alpha-1})^*\right]^*$, and its $\alpha$-controllability Gramian is $\mathcal{G}_{\mathrm{con}} = [R_1B, N_1R_1B, \ldots, N_1^{\alpha-1}R_1B]$. Per Lemma H.1 and Assumption 5.1, we know the $N_1, R_1B, CQ_1$ subsystem is both controllable and observable, and hence we can select $\alpha$ to be the smallest integer such that both $\mathcal{G}_{\mathrm{ob}}, \mathcal{G}_{\mathrm{con}}$ are rank-$k$. Note that we always have $\alpha \leq k$. Our main theorem will use the following system-theoretic parameters: $\alpha, \kappa(\mathcal{G}_{\mathrm{ob}}), \kappa(\mathcal{G}_{\mathrm{con}})$ (the condition numbers of $\mathcal{G}_{\mathrm{ob}}, \mathcal{G}_{\mathrm{con}}$ respectively), and $\sigma_{\min}(\mathcal{G}_{\mathrm{ob}}), \sigma_{\min}(\mathcal{G}_{\mathrm{con}})$ (the smallest singular values of $\mathcal{G}_{\mathrm{ob}}, \mathcal{G}_{\mathrm{con}}$, respectively).

**Umbrella upper bound $L$.** We use a constant $L$ to upper bound the norms $\|A\|, \|B\|, \|C\|, \|N_1\|, \|R_1B\|, \|CQ_1\|$. We will use the Jordan form decomposition for $N_1 = P_1\Lambda P_1^{-1}$, and let $L$ upper bound $\|P_1\|\|P_1^{-1}\| = \kappa(P_1)$. We also use $L$ in $\sup_{|z|=1}\|(zI-N_1)^{-1}\| \leq \frac{L}{|\lambda_k|-1}$. Lastly, we use $L$ to upper bound the constant in Gelfand's formula (Lemma H.2) for $N_2$ and $N_1^{-1}$. Specifically we will use $\|N_2^t\| \leq L\left(\frac{|\lambda_{k+1}|+1}{2}\right)^t, \|(N_1^{-1})^t\| \leq L\left(\frac{\frac{1}{|\lambda_k|}+1}{2}\right)^t, \forall t \geq 0$.

With the above preparations, we are ready to state the main theorem.

**Theorem 5.3.** *Suppose Assumption 5.1, Assumption 5.2 holds and $N_1$ is diagonalizable. In the regime where $|\lambda_k| - 1$, $1 - |\lambda_{k+1}|$, $\epsilon_*$ are small,[2] we set (recall throughout the paper $\asymp$ only hides numerical constants)*

$$m \asymp \max\left(\alpha, \frac{\log L}{1-|\lambda_{k+1}|}, \frac{1}{|\lambda_k|-1}\log\frac{\kappa(\mathcal{G}_{\mathrm{ob}})\kappa(\mathcal{G}_{\mathrm{con}})L}{\min(\sigma_{\min}(\mathcal{G}_{\mathrm{ob}}),\sigma_{\min}(\mathcal{G}_{\mathrm{con}}))(|\lambda_k|-1)\epsilon_*}\right),$$

*and we set $p = q = m$ (hence each trajectory's length is $T = 3m + 2$) and the number of trajectories*

$$M \asymp (\frac{\sigma_v}{\sigma_u})^2(d_u + d_y)\log\frac{m}{\delta} + d_um.$$

*for some $\delta \in (0, 1)$. Then, with probability at least $1 - \delta$, $\hat{K}(z)$ obtained by Algorithm 1 stabilizes the original dynamical system* (2).

---

[2] This regime is the most interesting and challenging regime. For more general values of $|\lambda_k| - 1$, $1 - |\lambda_{k+1}|$, $\epsilon_*$, see the bound in (106) which takes a more complicated form.

The total number of samples needed for the algorithm is $MT = (3m + 2)M$. In the bound in Theorem 5.3, the only constant that explicitly grows with $k$ is the controllability/observability index $\alpha = O(k)$ which appears in the bound for $m$. Therefore, the sample complexity $MT = O(m^2) = O(\alpha^2) = O(k^2)$ grows quadratically with $k$ and independent from system state dimension $n$. In the regime $k \ll n$, this significantly reduces the sample complexity of stabilization compared to methods that estimate the full system dynamics. To the best of our knowledge, this is the *first result that achieves stabilization of an unknown partially observable LTI system with sample complexity independent from the system state dimension $n$*. To validate the claim in Theorem 5.3, we use the LTS-P to stabilize a randomly generated partially observable system and compare the result with two state-of-the-art benchmarks [43, 51] in Appendix A. In Appendix B, we provide a high-level overview of the proof of Theorem 5.3, with detailed proof steps in Appendices C to E.

**Dependence on system theoretic parameters.** As the system becomes less observable and controllable, $\kappa(\mathcal{G}_{\mathrm{con}}), \kappa(\mathcal{G}_{\mathrm{ob}})$ increases and $\sigma_{\min}(\mathcal{G}_{\mathrm{con}}), \sigma_{\min}(\mathcal{G}_{\mathrm{ob}})$ decreases, increasing $m$ and the sample complexity $MT$ grows in the order of $(\log \frac{\kappa(\mathcal{G}_{\mathrm{ob}})\kappa(\mathcal{G}_{\mathrm{con}})}{\min(\sigma_{\min}(\mathcal{G}_{\mathrm{ob}}), \sigma_{\min}(\mathcal{G}_{\mathrm{con}}))})^2$. Moreover, when $|\lambda_{k+1}|, |\lambda_k|$ are closer to 1, the sample complexity also increases in the order of $(\max(\frac{1}{1-|\lambda_{k+1}|}, \frac{1}{|\lambda_k|-1} \log \frac{1}{|\lambda_k|-1}))^2$. This makes sense as the unstable and stable components of the system would become closer to marginal stability and harder to distinguish apart. Lastly, if the $\epsilon_*$ in Assumption 5.2 becomes smaller, we have a smaller margin for stabilization, which also increases the sample complexity in the order of $(\log \frac{1}{\epsilon_*})^2$.

**Non-diagonalization case.** Theorem 5.3 assumes $N_1$ is diagonalizable, which is only used in a single place in the proof in Lemma D.1. More specifically, we need to upper bound $J^t(J^*)^{-t}$ for some Jordan block $J$ and integer $t$, and in the diagonalizable case an upper bound is 1 as the Jordan block is size 1. In the case that $N_1$ is not diagonalizable, a similar bound can still be proven with dependence on the size of the largest Jordan block of $N_1$, denoted as $\gamma$. Eventually, this will lead to an additional multiplicative factor $\gamma$ in the sample complexity on $m$ (cf. (124)), whereas the requirements for $p, q, M$ is the same. The derivations can be found in Appendix G.

**Necessity of Assumption 5.2.** Assumption 5.2 is needed if one only estimates the unstable component of the system, treating the stable component as unknown. This is because the small gain theorem is an if-and-only-if condition. If Assumption 5.2 is not true, then no matter what controller $K(z)$ one designs, there must exist an adversarially chosen stable component $\Delta(z)$ (not known to the learner) causing the system to be unstable, even if $F(z)$ is perfectly learned. For a specific construction of such a stability-breaking $\Delta(z)$, see the proof of the small gain theorem, e.g. Theorem 8.1 of Zhou and Doyle [53]. Although Hu et al. [19], Zhang et al. [50] do not explictly impose this assumption, they impose similar assumptions on the eigenvalues, e.g. the $|\lambda_1||\lambda_{k+1}| < 1$ in Theorem 4.2 of Zhang et al. [50], and we believe the small gain theorem and Assumption 5.2 is the intrinsic reason underlying those eigenvalue assumptions in Zhang et al. [50]. We believe this requirement of Assumption 5.2 is the fundamental limit of the unstable + stable decomposition approach. One way to break this fundamental limit is that instead of doing unstable + stable decomposition, we learn a larger component corresponding to eigenvalues $\lambda_1, \ldots, \lambda_{\tilde{k}}$ for some $\tilde{k} > k$, which effectively means we learn the unstable component and part of the stable component of the system. Doing so, Assumption 5.2 will be easier to satisfy as when $\tilde{k}$ approaches $n$, $\|\Delta(z)\|_{H_\infty}$ will approach 0.

**Relation to hardness of control results.** Recently, there has been related work on the hardness of the learn-to-stabilize problem [9, 48]. Our setting does not conflict with these hardness results for the following reasons. Firstly, our result focuses on the $k \ll n$ regime. In the regime $k = n$, our result does not improve over other approaches. Secondly, our Assumption 5.2 is related to the co-stabilizability concept in Zeng et al. [48], as Assumption 5.2 effectively means the existence of a controller that stabilizes the system for all possible $\Delta(z)$. In some sense, our results complements these results showing when is learn-to-stabilize easy under certain assumptions.

# 6 Conclusion

In this paper, we examined the necessary and sufficient conditions for the stabilization of a partially observable LTI system and proposed a novel SVD-based robust controller design framework. Future work includes studying the transfer function from the process noise $w_t$ to $y_t$, and improving the complexity bound in Theorem 5.3.

# 7 Acknowledgement

This work is supported in part by NSF 2154171, NSF CAREER 2339112, CMU CyLab seed funding, in part by by a grant from the Commonwealth of Pennsylvania, Department of Community and Economic Development, in part by National Science Foundation under Grant No. 2442948, and in part by the Department of the Navy, Office of Naval Research, under award number N00014-23-1-2252. The views expressed are those of the authors and do not reflect the official policy or position of the US Navy, Department of Defense or the US Government.

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

# A   Simulations

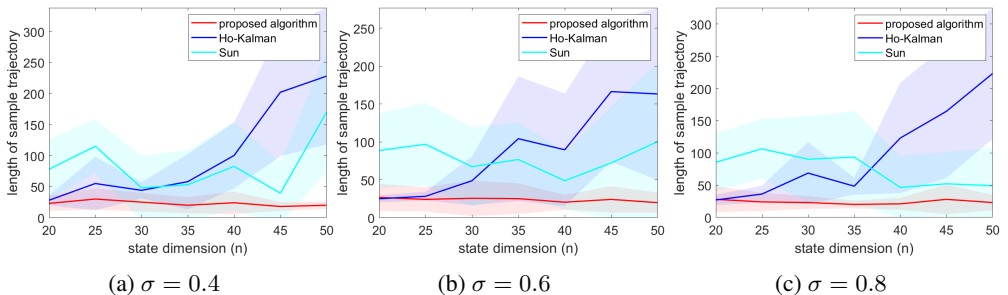

(a) $\sigma = 0.4$          (b) $\sigma = 0.6$          (c) $\sigma = 0.8$

Figure 1: The above figure shows the length of rollouts needed to identify and stabilize an unstable system with the unstable dimension $k = 5$. The solid line shows the average length of rollouts the learner takes to stabilize the system. The shaded area shows the standard deviation of the length of rollouts. The proposed method requires the shortest rollouts and has the smallest standard deviation.

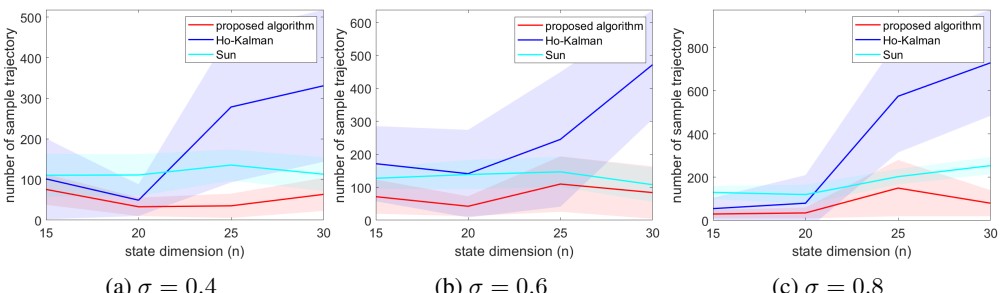

(a) $\sigma = 0.4$          (b) $\sigma = 0.6$          (c) $\sigma = 0.8$

Figure 2: The above figure shows the number of rollouts needed to identify and stabilize an unstable system with the unstable dimension $k = 5$. The solid line shows the average number of rollouts the learner takes to stabilize the system. The shaded area shows the standard deviation of the number of rollouts. The proposed method requires the least number rollouts and has small standard deviation.

In this section, we compare our algorithm to the two existing benchmarks: nuclear norm regularization in Sun et al. [43] and the Ho-Kalman algorithm in Zheng and Li [51]. We use a more complicated system than (1):

$$
\begin{aligned}
x_{t+1} &= Ax_t + Bu_t + w_t \\
y_t &= Cx_t + Du_t + v_t,
\end{aligned}
\tag{26}
$$

where there are both process noise $w_t$ and observation noise $v_t$. we fix the unstable part of the LTI system to dimension 5, i.e. $k = 5$, and observe the effect of increasing dimension $n$ on the system. For each dimension $n$, we randomly generate a matrix with $k$ unstable eigenvalues $\lambda_i \sim \text{unif}(1.1, 2)$

and $n - k$ stable eigenvalues $\lambda_j \sim \mathrm{unif}(0, 0.5)$. The basis for these eigenvalues is also initialized randomly. In both parts of the simulations, we fix $m = 4$ and $p = q = \lfloor \frac{T-4}{2} \rfloor$ for the proposed algorithm, LTS-P.

In the first part of the simulations, we fix the number of rollouts to be $4$ times the dimension of the system and let the system run with an increasing length of rollouts until a stabilizing controller is generated. The system is simulated in three different settings, each with noise $w_t, v_t \sim \mathcal{N}(0, \sigma)$, with $\sigma = 0.4, 0.6, 0.8$. For each of the three algorithms, the simulation is repeated for 30 trials, and the result is shown in Figure 1. The proposed method requires the shortest rollouts. The algorithm in Sun et al. [43] roughly have the same length of rollouts across different dimensions. This is to be expected, as Sun et al. [43] only uses the last data point of each rollout, so a longer trajectory does not generate more data. However, unlike the proposed algorithm, Sun et al. [43] still needs to estimate the entire system, which is why it still needs a longer rollout. The length of the rollouts of Zheng and Li [51] grows linearly with the dimension of the system. We also see fluctuations of the length of rollouts required for stabilization in Figure 1 because the matrix generated for each $n$ has a randomly generated eigenbasis, which influences the amount of data needed for stabilization, as shown in Theorem 5.3. Overall, this simulation shows that the length of rollouts of the proposed algorithm remains $O(\mathrm{poly}(k))$ regardless of the dimension of the system $n$.

In the second part of the simulation, we fix the length of each trajectory to $70$ and examine the number of trajectories needed before the system stabilizes. We do not increase the length of trajectory in this part of the simulation with increasing $n$, since Sun et al. [43] only uses the last data point, and does not show significant performance improvement with longer trajectories, as shown in the first part of the simulation, so it would be an unfair comparison if the trajectory is overly long. Similar to the first part, the system is simulated in three different settings and the result is shown in Figure 2. Overall, the proposed method requires the least number of trajectories. When the $n \gtrsim k$, the proposed algorithm requires a similar number of data with Zheng and Li [51], as the proposed algorithm is the same to that in Zheng and Li [51] when $k = n$. When $n \gg k$, the proposed algorithm outperforms both benchmarks.

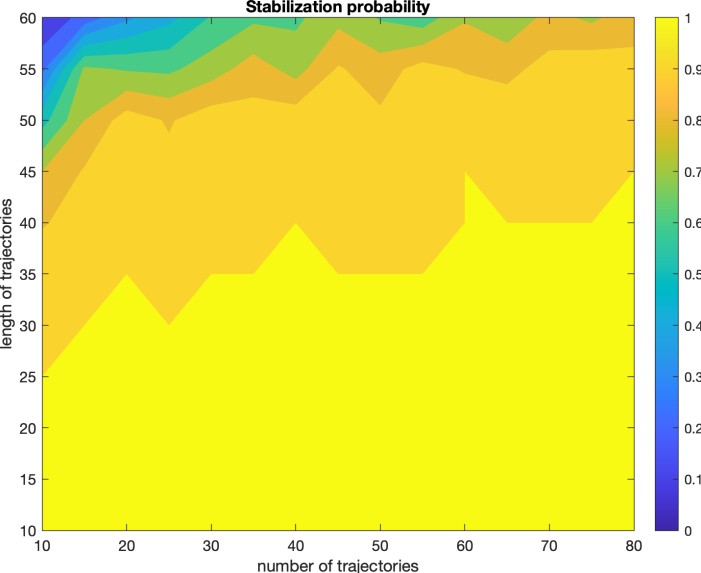

Figure 3: The above figure shows the probability of stabilization for a randomly generated matrix with $n = 15$ and $k = 5$.

In the last simulation, we fix a randomly generated matrix with dimension $n = 15$ and an unstable subspace of dimension $k = 5$. We further set the variance of the process and observation noise to $\sigma = 0.4$. As shown in Figure 3, the stabilization probability is most enhanced by adding more trajectories, but longer trajectories also demonstrate a significant effect for stabilization, as proven in Theorem 5.3.

# B  Proof Outline

In this section, we will give a high-level overview of the key proof ideas for the main theorem. The full proof details can be found in Appendix C (for Step 1), Appendix D (for Step 2), and Appendix E (for Step 3).

**Proof Structure.**  The proof is largely divided into three steps following the idea developed in Section 3. In the first step, we examine how accurately the learner can estimate the low-rank lifted Hankel matrix $\tilde{H}$. In the second step, we examine how accurately the learner estimates the transfer function of the unstable component from $\tilde{H}$. In the last step, we provide a stability guarantee.

**Step 1.**  We show that $\hat{\tilde{H}}$ obtained in (19) is an accurate estimate of $\tilde{H}$.

**Lemma B.1.**  *With probability at least $1 - \delta$, the estimation $\hat{\tilde{H}}$ obtained in (19) satisfies*

$$\|\Delta_H\| := \left\| \hat{\tilde{H}} - \tilde{H} \right\| < \epsilon := 2\hat{\epsilon} + 2\tilde{\epsilon},$$

*where $\hat{\epsilon}$ is the estimation error for $\|H - \hat{H}\|$ that decays in the order of $O(\frac{1}{\sqrt{M}})$; $\tilde{\epsilon}$ is the error $\left\| H - \tilde{H} \right\|$ that decays in the order of $O((\frac{|\lambda_{k+1}|+1}{2})^m)$. See Theorem C.2 and Lemma C.3 for the exact constants.*

We later (in Step 2, 3) will show that for a sufficiently small $\epsilon$, the robust controller produced by the algorithm will stabilize the system in (1). We will invoke Lemma B.1 at the end of the proof in Step 3 to pick $m, p, q, M$ values that achieve the required $\epsilon$ (in Appendix E).

**Step 2.**  In this step, we will show that $N_1, R_1 B, C Q_1$ can be estimated up to an error of $O(\epsilon)$ and up to a similarity transformation, given that $\left\| \hat{\tilde{H}} - \tilde{H} \right\| < \epsilon$.

**Lemma B.2.**  *The estimated system dynamics $\hat{N}_1$ satisfies the following bound for some invertible matrix $\hat{S}$:*

$$\|\hat{S}^{-1}\hat{N}_1\hat{S} - N_1\| \leq \frac{4 L c_2 \epsilon}{c_1 \tilde{o}_{\min}(m)} := \epsilon_N.$$

*where $c_2, c_1$ are constants depending on system theoretic parameters, and $\tilde{o}_{\min}(m)$ is a quantity that grows with $m$. See Lemma D.1 for the definition of these quantities.*

**Lemma B.3.**  *Up to transformation $\hat{S}$ (same as the one in Lemma B.2), we can bound the difference between $\hat{C}$ and the unstable component of the system $C Q_1$ as follows:*

$$\left\| C Q_1 - \hat{C}\hat{S} \right\| \leq 2 \left( \frac{4 c_2 L}{c_1 \tilde{o}_{\min}(m)} + \frac{1}{\tilde{c}_{\min}(m)} \right) \epsilon := \epsilon_C.$$

*where $\tilde{c}_{\min}(m)$ is a quantity that grows with $m$ defined in Lemma D.1.*

**Lemma B.4.**  *Up to transformation $\hat{S}^{-1}$ ($\hat{S}$ is the same as the one in Lemma B.2), we can bound the difference between $\hat{B}$ and the unstable component of the system $R_1 B$ as follows:*

$$\left\| R_1 B - \hat{S}^{-1}\hat{B} \right\| < 8 \left( \frac{c_2 L}{c_1 \tilde{c}_{\min}(m)} + \frac{1}{\tilde{o}_{\min}(m)} \right) \epsilon := \epsilon_B.$$

**Step 3.**  We show that, under sufficiently accurate estimation, the controller $\hat{K}(z)$ returned by the algorithm stabilizes plant $F(z)$ and hence $\|\hat{K}(z)F_{\hat{K}}(z)\|_{H_\infty}$ is well defined. This is done via the following helper proposition.

**Proposition B.5.**  *Recall the definition of the estimated transfer function in (25). Suppose the estimation errors $\epsilon_N, \epsilon_C, \epsilon_B$ are small enough such that $\sup_{|z|=1} \left\| F(z) - \hat{F}(z) \right\| < \epsilon_*$. Then, the following two statements hold: (a) If $K(z)$ stabilizes $F(z)$ and $\|K(z)F_K(z)\|_{H_\infty} < \frac{1}{\epsilon_*}$, then $K(z)$ stabilizes plant $\hat{F}(z)$ as well; (b) similarly, if $\hat{K}(z)$ stabilizes $\hat{F}(z)$ with $\|\hat{K}(z)\hat{F}_{\hat{K}}(z)\|_{H_\infty} < \frac{1}{\epsilon_*}$, then $\hat{K}(z)$ stabilizes plant $F(z)$ as well.*

Then, we upper bound $\|\hat{K}(z)F_{\hat{K}}(z)\|_{H_\infty}$ and use small gain theorem (Lemma 3.1) to show that the stable mode $\Delta(z)$ does not affect stability either, and therefore, the controller $\hat{K}(z)$ stabilizes the full system, $F^{\text{full}}(z)$ in (4). Leveraging the error bounds in Step 2 and Step 1, we will also provide the final sample complexity bound. The detailed proof can be found in Appendix E.

## C Proof of Step 1: Bounding Hankel Matrix Error

To further simplify notation, define

$$Y = \begin{bmatrix} y^{(1)} & \dots & y^{(M)} \end{bmatrix}, \tag{27}$$

$$U = \begin{bmatrix} U^{(1)} & \dots & U^{(M)} \end{bmatrix}, \tag{28}$$

$$v = \begin{bmatrix} v^{(1)} & \dots & v^{(M)} \end{bmatrix}, \tag{29}$$

so (17) can be simplified to

$$\hat{\Phi} = \arg \min_{X \in \mathbb{R}^{d_y \times (T*d_u)}} \|Y - XU\|_F^2, \tag{30}$$

respectively. An analytic solution to (30) is $\hat{\Phi} := YU^\dagger$, where $U^\dagger := U^*(UU^*)^{-1}$ is the right psuedo-inverse of $U$. In this paper, we will use $\hat{\tilde{\Phi}}$ to recover $\tilde{H}$, and we want to bound the error between $\tilde{H}$ and $\hat{\tilde{H}}$.

$$\begin{aligned} \hat{\Phi} - \Phi &= YU^*(UU^*)^{-1} - \Phi \\ &= (Y - \Phi U)U^*(UU^*)^{-1} \\ &= vU^*(UU^*)^{-1}, \end{aligned} \tag{31}$$

where the third equality used (16). We can bound the estimation error (31) by

$$\left\| \hat{\Phi} - \Phi \right\| \le \left\| (UU^*)^{-1} \right\| \left\| vU^* \right\|. \tag{32}$$

We then prove a lemma that links the difference between $\Phi$ and Hankel matrix $H$.

**Lemma C.1.** *Given two Hankel matrices $H^\clubsuit, H^\diamondsuit$ constructed from $\Phi^\clubsuit, \Phi^\diamondsuit$ with (18), respectively, then*

$$\left\| H^\clubsuit - H^\diamondsuit \right\| \le \sqrt{\min\{p, q\}} \left\| \Phi^\clubsuit_{2+m:T} - \Phi^\diamondsuit_{2+m:T} \right\|.$$

*Proof.* From the construction in (18), we see that each column of $H^\clubsuit - H^\diamondsuit$ is a submatrix of $\Phi^\clubsuit_{2+m:T} - \Phi^\diamondsuit_{2+m:T}$, then

$$\left\| H^\clubsuit - H^\diamondsuit \right\| \le \sqrt{q} \left\| \Phi^\clubsuit_{2+m:T} - \Phi^\diamondsuit_{2+m:T} \right\|. \tag{33}$$

Similarly, each row of $H^\clubsuit - H^\diamondsuit$ is a submatrix of $\Phi^\clubsuit_{2+m:T} - \Phi^\diamondsuit_{2+m:T}$. Therefore,

$$\left\| H^\clubsuit - H^\diamondsuit \right\| \le \sqrt{p} \left\| \Phi^\clubsuit_{2+m:T} - \Phi^\diamondsuit_{2+m:T} \right\|. \tag{34}$$

Combining (33) and (34) gives the desired result. □

We then bound the gap between $H^{(k)}$ and $\hat{\tilde{H}}$ by adapting Theorem 3.1 of Zheng and Li [51]. For completeness, we put the theorem and proof below:

**Theorem C.2.** *For any $\delta > 0$, the following inequality holds when $M > 8d_uT + 4(d_u + d_y + 4)\log(3T/\delta)$ with probability at least $1 - \delta$:*

$$\left\| H - \hat{H} \right\| \le \frac{8\sigma_v \sqrt{T(T+1)(d_u + d_y)\min\{p, q\}\log(27T/\delta)}}{\sigma_u\sqrt{M}} := \hat{\epsilon}$$

*Proof.* By Lemma C.1, we have

$$\left\| H - \hat{H} \right\| \leq \sqrt{\min\{p, q\}} \left\| \Phi_{2+m:T} - \hat{\Phi}_{2+m:T} \right\|.$$

Bounding $\|(UU^*)\|^{-1}$ and $\|vU^*\|$ with Proposition 3.1 and Proposition 3.2 of [51] gives the result in the theorem statement. $\qquad\square$

**Lemma C.3.**

$$\left\| H - \tilde{H} \right\| \leq \sqrt{q}p \leq L^3 \left( \frac{|\lambda_{k+1}| + 1}{2} \right)^m := \tilde{\epsilon}$$

*Proof.* We first bound $H$ and $\tilde{H}$ entry-wise,

$$
\begin{aligned}
H_{i,j} - \tilde{H}_{i,j} &= CA^{m+i+j-2}B - CQ_1 N_1^{m+i+j-2} R_1 B \\
&= C(Q_1 N_1^{m+i+j-2} R_1 + Q_2 N_2^{m+i+j-2} R_2)B - CQ_1 N_1^{m+i+j-2} R_1 B \\
&= CQ_2 N_2^{m+i+j-2} R_2 B.
\end{aligned}
$$

Applying Gelfand's formula (Lemma H.2), we obtain

$$\left\| H_{i,j} - \tilde{H}_{i,j} \right\| \leq \|B\| \|C\| \zeta_{\epsilon_2}(N_2)(|\lambda_{k+1}| + \epsilon_4)^{m+i+j-2} \leq L^3 \left( \frac{|\lambda_{k+1}| + 1}{2} \right)^{m+i+j-2}. \quad (35)$$

Moreover, we have the following well-known inequality (see, for example, [17]), for $\Lambda \in \mathbb{R}^{n_1 \times n_2}$,

$$\frac{1}{\sqrt{n_1}} \|\Lambda\|_1 \leq \|\Lambda\|_2 \leq \sqrt{n_2} \|\Lambda\|_1. \quad (36)$$

Applying (36) to (35) gives us the desired inequality. $\qquad\square$

We are now ready to prove Lemma B.1.

*Proof of Lemma B.1.* We first provide a bound on $\hat{\tilde{H}} - \hat{H}$. As $\hat{\tilde{H}}$ is the rank-$k$ approximation of $\hat{H}$, we have $\left\| \hat{\tilde{H}} - \hat{H} \right\| \leq \left\| \tilde{H} - \hat{H} \right\|$, then

$$\left\| \hat{\tilde{H}} - \hat{H} \right\| \leq \left\| \tilde{H} - H \right\| + \left\| H - \hat{H} \right\| = \tilde{\epsilon} + \hat{\epsilon}. \quad (37)$$

where we use Theorem C.2 and Lemma C.3. Therefore, we obtain the following bound

$$\left\| \hat{\tilde{H}} - \tilde{H} \right\| \leq \left\| \hat{\tilde{H}} - \hat{H} \right\| + \left\| \hat{H} - H \right\| + \left\| H - \tilde{H} \right\| \leq 2\hat{\epsilon} + 2\tilde{\epsilon} := \epsilon. \quad (38)$$

$\qquad\square$

# D   Proof of Step 2: Bounding Dynamics Error

In the previous section, we proved Lemma B.1 and provided a bound for $\|\hat{\tilde{H}} - \tilde{H}\|$. In this section, we use this bound to derive the error bound for our system estimation. We start with the following lemma that bounds the difference between $\tilde{\mathcal{O}}$ and $\hat{\mathcal{O}}$ and the difference between $\tilde{\mathcal{C}}$ and $\hat{\mathcal{C}}$ up to a transformation.

**Lemma D.1.** *Given $N_1$ is diagnalizable, and recall $N_1 = P_1 \Lambda P_1^{-1}$ denotes the eigenvalue decomposition of $N_1$, where $\Lambda = \mathrm{diag}(\lambda_1, \ldots, \lambda_k)$ is the diagonal matrix of eigenvalues. Consider $\hat{\tilde{H}} = \hat{\mathcal{O}}\hat{\mathcal{C}}$ and $\tilde{H} = \tilde{\mathcal{O}}\tilde{\mathcal{C}}$, where we recall $\hat{\mathcal{O}}, \hat{\mathcal{C}}$ are obtained by doing SVD $\hat{\tilde{H}} = \hat{U}\hat{\Sigma}\hat{V}^*$ and setting $\hat{\mathcal{O}} = \hat{U}\hat{\Sigma}^{1/2}, \hat{\mathcal{C}} = \hat{\Sigma}^{1/2}\hat{V}^*$ (cf. (20)). Suppose $\|\Delta_H\| = \left\| \tilde{H} - \hat{\tilde{H}} \right\| \leq \epsilon$, then there exists an invertible $k$-by-$k$ matrix $\hat{S}$ such that*

*(a)* $\left\| \tilde{\mathcal{O}} - \hat{\mathcal{O}}\hat{S} \right\| \leq \frac{1}{\sigma_{\min}(\tilde{\mathcal{C}})}\epsilon$ *and* $\left\| \tilde{\mathcal{C}} - \hat{S}^{-1}\hat{\mathcal{C}} \right\| \leq \frac{4}{\sigma_{\min}(\tilde{\mathcal{O}})}\epsilon$,

*(b) there exists $c_1, c_2 > 0$ such that $c_1 I \preceq \hat{S} \preceq c_2 I$, where given $\underline{\sigma}_s, \overline{\sigma}_s$ in (49),*

$$c_1^2 = \frac{\underline{\sigma}_s}{2\sqrt{2}\sigma_{\max}(P_1)^2}, \qquad c_2^2 = 2\sqrt{2}\frac{\overline{\sigma}_s}{\sigma_{\min}(P_1)^2}.$$

*(c) we can bound $\sigma_{\min}(\tilde{\mathcal{O}})$ and $\sigma_{\min}(\tilde{\mathcal{C}})$ as follows:*

$$\sigma_{\min}(\tilde{\mathcal{O}}) \geq \sigma_{\min}(\tilde{\mathcal{O}}_{1:\alpha}) \geq \sigma_{\min}(\mathcal{G}_{\mathrm{ob}})\sigma_{\min}(N_1^{m/2}) := \tilde{o}_{\min}(m),$$

$$\sigma_{\min}(\tilde{\mathcal{C}}) \geq \sigma_{\min}(\tilde{\mathcal{C}}_{1:\alpha}) \geq \sigma_{\min}(\mathcal{G}_{\mathrm{con}})\sigma_{\min}(N_1^{m/2}) := \tilde{c}_{\min}(m).$$

*Proof.* **Proof of part(a)**. Recall the definition of $\tilde{H}$

$$\tilde{H} = [CQ_1 N_1^{m+i+j} R_1 B]_{i\in\{0,\dots,p-1\}, j\in\{0,\dots,q-1\}}$$

$$= \underbrace{\begin{bmatrix} CQ_1 N_1^{\frac{m}{2}} \\ \dots \\ CQ_1 N_1^{\frac{m}{2}+p-1} \end{bmatrix}}_{\tilde{\mathcal{O}}} \underbrace{\begin{bmatrix} N_1^{\frac{m}{2}} R_1 B & \dots & N_1^{\frac{m}{2}+q-1} R_1 B \end{bmatrix}}_{\tilde{\mathcal{C}}}.$$

We can represent $\tilde{\mathcal{O}}$ and $\tilde{\mathcal{C}}$ as follows:

$$\tilde{\mathcal{O}} = \underbrace{\begin{bmatrix} CQ_1 P_1 & & & \\ & CQ_1 P_1 & & \\ & & \ddots & \\ & & & CQ_1 P_1 \end{bmatrix} \begin{bmatrix} \Lambda^{\frac{m}{2}} & & \\ & \ddots & \\ & & \Lambda^{\frac{m}{2}+p-1} \end{bmatrix}}_{:=G_1} P_1^{-1}. \tag{39}$$

$$\tilde{\mathcal{C}} = P_1 \underbrace{\begin{bmatrix} \Lambda^{\frac{m}{2}} & \dots & \Lambda^{\frac{m}{2}+q-1} \end{bmatrix} \begin{bmatrix} P_1^{-1} R_1 B & & \\ & \ddots & \\ & & P_1^{-1} R_1 B \end{bmatrix}}_{:=G_2^*}. \tag{40}$$

Next, note that

$$\hat{\tilde{H}} = \hat{U}\hat{\Sigma}\hat{V}^* = \hat{U}\hat{\Sigma}^{\frac{1}{2}}\hat{\Sigma}^{\frac{1}{2}}\hat{V}^* = \hat{\mathcal{O}}\hat{\mathcal{C}} = \tilde{\mathcal{O}}\tilde{\mathcal{C}} + \Delta_H = G_1 G_2^* + \Delta_H. \tag{41}$$

Therefore, we have

$$G_1 G_2^* G_2 = \underbrace{\hat{U}\hat{\Sigma}^{\frac{1}{2}}}_{\hat{\mathcal{O}}}\hat{\Sigma}^{\frac{1}{2}}\hat{V}^* G_2 - \Delta_H G_2$$

$$\Rightarrow G_1 = \hat{\mathcal{O}}\hat{\Sigma}^{\frac{1}{2}}\hat{V}^* G_2(G_2^* G_2)^{-1} - \Delta_H G_2(G_2^* G_2)^{-1} \tag{42}$$

$$\Rightarrow \tilde{\mathcal{O}} = G_1 P_1^{-1} = \hat{\mathcal{O}}\underbrace{\hat{\Sigma}^{\frac{1}{2}}\hat{V}^* G_2(G_2^* G_2)^{-1} P_1^{-1}}_{:=\hat{S}} - \Delta_H G_2(G_2^* G_2)^{-1} P_1^{-1}.$$

where we have defined the matrix $\hat{S}$ above. We can also expand $\hat{S}$ as follows:

$$\hat{S} = \hat{\Sigma}^{\frac{1}{2}}\hat{V}^* G_2 P_1^*(P_1^*)^{-1}(G_2^* G_2)^{-1} P_1^{-1}$$

$$= \hat{\Sigma}^{\frac{1}{2}}\hat{V}^* G_2 P_1^*(P_1 G_2^* G_2 P_1^*)^{-1}$$

$$= \hat{\mathcal{C}}\tilde{\mathcal{C}}^*(\tilde{\mathcal{C}}\tilde{\mathcal{C}}^*)^{-1}. \tag{43}$$

Further, the last term in the last equation in (42) can be written as,

$$\Delta_H G_2(G_2^* G_2)^{-1} P_1^{-1} = \Delta_H G_2 P_1^*(P_1^*)^{-1}(G_2^* G_2)^{-1} P_1^{-1}$$

$$= \Delta_H \tilde{\mathcal{C}}^*(\tilde{\mathcal{C}}\tilde{\mathcal{C}}^*)^{-1}. \tag{44}$$

Putting (42), (44) together, we have

$$\left\|\tilde{\mathcal{O}} - \hat{\mathcal{O}}\hat{S}\right\| \le \frac{1}{\sigma_{\min}(\tilde{\mathcal{C}})}\epsilon, \tag{45}$$

which proves one side of part (a). To finish part(a), we are left to bound $\left\|\tilde{\mathcal{C}} - \hat{S}^{-1}\hat{\mathcal{C}}\right\|$. With the expression of $\hat{S}$ in (43), we obtain

$$\left\|\hat{\mathcal{C}} - \hat{S}^{-1}\tilde{\mathcal{C}}\right\| = \left\|\hat{\mathcal{C}} - \tilde{\mathcal{C}}\tilde{\mathcal{C}}^*(\hat{\mathcal{C}}\tilde{\mathcal{C}}^*)^{-1}\tilde{\mathcal{C}}\right\|.$$

Let $E := \hat{\mathcal{O}}\hat{S} - \tilde{\mathcal{O}}$ so $\tilde{\mathcal{O}} = \hat{\mathcal{O}}\hat{S} - E$. By (42) and (44), we obtain

$$E = \Delta_H \tilde{\mathcal{C}}^*(\tilde{\mathcal{C}}\tilde{\mathcal{C}}^*)^{-1}, \tag{46}$$

and

$$\left\|E\tilde{\mathcal{C}}\right\| = \left\|\Delta_H \tilde{\mathcal{C}}^*(\tilde{\mathcal{C}}\tilde{\mathcal{C}}^*)^{-1}\tilde{\mathcal{C}}\right\| \le \epsilon.$$

Therefore, we can obtain the following inequalities:

$$\epsilon \ge \left\|\tilde{\mathcal{O}}\tilde{\mathcal{C}} - \hat{\mathcal{O}}\hat{\mathcal{C}}\right\| = \left\|(\hat{\mathcal{O}}\hat{S} - E)\tilde{\mathcal{C}} - \hat{\mathcal{O}}\hat{\mathcal{C}}\right\| = \left\|\hat{\mathcal{O}}(\hat{S}\tilde{\mathcal{C}} - \hat{\mathcal{C}}) - E\tilde{\mathcal{C}}\right\| \ge \left\|\hat{\mathcal{O}}(\hat{S}\tilde{\mathcal{C}} - \hat{\mathcal{C}})\right\| - \left\|E\tilde{\mathcal{C}}\right\|$$

$$\Rightarrow \left\|\hat{\mathcal{O}}\hat{S}(\tilde{\mathcal{C}} - \hat{S}^{-1}\hat{\mathcal{C}})\right\| \le 2\epsilon$$

$$\Rightarrow \left\|\tilde{\mathcal{C}} - \hat{S}^{-1}\hat{\mathcal{C}}\right\| \le \frac{2}{\sigma_{\min}(\hat{\mathcal{O}}\hat{S})}\epsilon \le \frac{2}{\sigma_{\min}(\tilde{\mathcal{O}}) - \left\|\tilde{\mathcal{O}} - \hat{\mathcal{O}}\hat{S}\right\|}\epsilon.$$

Substituting (45), we get

$$\left\|\tilde{\mathcal{C}} - \hat{S}^{-1}\hat{\mathcal{C}}\right\| \le \frac{2\epsilon}{\sigma_{\min}(\tilde{\mathcal{O}}) - \frac{1}{\sigma_{\min}(\tilde{\mathcal{C}})}\epsilon} \le \frac{4}{\sigma_{\min}(\tilde{\mathcal{O}})}\epsilon.$$

where the last inequality requires

$$\epsilon < \frac{1}{2}\sigma_{\min}(\tilde{\mathcal{O}})\sigma_{\min}(\tilde{\mathcal{C}}). \tag{47}$$

which we will verify at the end of the proof of Theorem 5.3 that our selection of algorithm parameters guarantees is true.

**Proof of part(b)**. We first examine the relationship between $G_1$ and $G_2$. Recall $\alpha$ is the maximum of the controllability and observability index. Therefore, the matrix

$$d_u * \alpha \left\{ \begin{bmatrix} \overbrace{B^* R_1^*(P_1^{-1})^*}^{k} \\ B^* R_1^*(P_1^{-1})^* \Lambda^* \\ \vdots \\ B^* R_1^*(P_1^{-1})^*(\Lambda^{\alpha-1})^* \end{bmatrix} \right., $$

which is exactly $\mathcal{G}_{\text{con}}^*(P_1^*)^{-1}$, is full column rank. For any full column rank matrix $\Gamma$, let's $\Gamma^\dagger = (\Gamma^*\Gamma)^{-1}\Gamma^*$ denote its Moore-Penrose inverse that satisfies $\Gamma^\dagger\Gamma = I$. With this, for non-negative integer $j$ we define the following matrix, which is essentially $[G_1]_{j\alpha:(j+1)\alpha-1}$ "divided by" $[G_2]_{j\alpha:(j+1)\alpha-1}$ in the Moore-Penrose sense:

$$S_{CB}^{(j)} = \begin{bmatrix} CQ_1 P_1 \Lambda^{\frac{m}{2}+j\alpha} \\ CQ_1 P_1 \Lambda^{\frac{m}{2}+j\alpha+1} \\ \vdots \\ CQ_1 P_1 \Lambda^{\frac{m}{2}+(j+1)\alpha-1} \end{bmatrix} \begin{bmatrix} B^* R_1^*(P_1^{-1})^*(\Lambda^{\frac{m}{2}+j\alpha})^* \\ B^* R_1^*(P_1^{-1})^*(\Lambda^{\frac{m}{2}+j\alpha+1})^* \\ \vdots \\ B^* R_1^*(P_1^{-1})^*(\Lambda^{\frac{m}{2}+(j+1)\alpha-1})^* \end{bmatrix}^\dagger$$

$$= \underbrace{\begin{bmatrix} CQ_1P_1 \\ CQ_1P_1\Lambda \\ \vdots \\ CQ_1P_1\Lambda^{\alpha-1} \end{bmatrix}}_{=\mathcal{G}_{\mathrm{ob}}P_1} \Lambda^{\frac{m}{2}+j\alpha}(\Lambda^{-\frac{m}{2}-j\alpha})^* \underbrace{\begin{bmatrix} B^*R_1^*(P_1^{-1})^* \\ B^*R_1^*(P_1^{-1})^*\Lambda^* \\ \vdots \\ B^*R_1^*(P_1^{-1})^*(\Lambda^{\alpha-1})^* \end{bmatrix}^\dagger}_{=(\mathcal{G}_{\mathrm{con}}^*(P_1^*)^{-1})^{-1}}. \tag{48}$$

As in the main Theorem 5.3 we assumed the $N_1$ is diagonalizable, $\Lambda$ is a diagonal matrix with possibly complex entries. As such, $\Lambda^{\frac{m}{2}+j\alpha}(\Lambda^{-\frac{m}{2}-j\alpha})^*$ is a diagonal matrix with all entries having modulus 1.[3] Therefore, we have

$$\sigma_{\min}(S_{CB}^{(j)}) \geq \frac{\sigma_{\min}(\mathcal{G}_{\mathrm{ob}})}{\sigma_{\max}(\mathcal{G}_{\mathrm{con}})}\sigma_{\min}(P_1)^2 := \underline{\sigma}_s,$$

$$\sigma_{\max}(S_{CB}^{(j)}) \leq \frac{\sigma_{\max}(\mathcal{G}_{\mathrm{ob}})}{\sigma_{\min}(\mathcal{G}_{\mathrm{con}})}\sigma_{\max}(P_1)^2 := \overline{\sigma}_s, \tag{49}$$

for all $j$. Recall the condition in the main theorem 5.3 requires $p = q$. Without loss of generality, we can assume $\alpha$ divides $p$ and $q$,[4] then $G_1$ and $G_2$ can be related in the following way:

$$G_1 = \underbrace{\begin{bmatrix} S_{CB}^{(0)} & & & \\ & S_{CB}^{(1)} & & \\ & & \ddots & \\ & & & S_{CB}^{(\frac{p}{\alpha})} \end{bmatrix}}_{:=\mathcal{S}_{CB}} G_2. \tag{50}$$

With the above relation, we are now ready to bound the norm of $\hat{S}$. We calculate

$$\hat{S}^*\hat{S} = (P_1^{-1})^*(G_2^*G_2)^{-1}G_2^*\hat{V}\hat{\Sigma}\hat{V}^*G_2(G_2^*G_2)^{-1}P_1^{-1}. \tag{51}$$

If we examine the middle term $\hat{V}\hat{\Sigma}\hat{V}^*$, we can expand it as

$$\begin{aligned}
\hat{V}\hat{\Sigma}\hat{V}^* &= (\hat{V}\hat{\Sigma}^2\hat{V}^*)^{\frac{1}{2}} \\
&= (\hat{V}\hat{\Sigma}\hat{U}^*\hat{U}\hat{\Sigma}\hat{V}^*)^{\frac{1}{2}} \\
&= (\hat{\hat{H}}^*\hat{\hat{H}})^{\frac{1}{2}} \\
&= ((\tilde{H}+\Delta_H)^*(\tilde{H}+\Delta_H))^{\frac{1}{2}}. \tag{52}
\end{aligned}$$

Using Lemma H.3 on (52), we obtain

$$(\frac{1}{2}\tilde{H}^*\tilde{H} - \Delta_H^*\Delta_H)^{\frac{1}{2}} \preceq \hat{V}\hat{\Sigma}\hat{V}^* \preceq (2\tilde{H}^*\tilde{H} + 2\Delta_H^*\Delta_H)^{\frac{1}{2}}$$

$$\Rightarrow \left(\frac{1}{2}(G_1G_2^*)^*G_1G_2^* - \Delta_H^*\Delta_H\right)^{\frac{1}{2}} \preceq \hat{V}\hat{\Sigma}\hat{V}^* \preceq (2(G_1G_2^*)^*G_1G_2^* + 2\Delta_H^*\Delta_H)^{\frac{1}{2}} \quad \text{subsitute}(41)$$

$$\Rightarrow \left(\frac{1}{2}G_2G_1^*G_1G_2^* - \Delta_H^*\Delta_H\right)^{\frac{1}{2}} \preceq \hat{V}\hat{\Sigma}\hat{V}^* \preceq (2G_2G_1^*G_1G_2^* + 2\Delta_H^*\Delta_H)^{\frac{1}{2}}$$

$$\Rightarrow \left(\frac{1}{2}G_2G_2^*\mathcal{S}_{CB}^*\mathcal{S}_{CB}G_2G_2^* - \Delta_H^*\Delta_H\right)^{\frac{1}{2}} \preceq \hat{V}\hat{\Sigma}\hat{V}^* \preceq (2G_2G_2^*\mathcal{S}_{CB}^*\mathcal{S}_{CB}G_2G_2^* + 2\Delta_H^*\Delta_H)^{\frac{1}{2}} \quad \text{substitute}(50).$$

Therefore, we can bound $\hat{V}\hat{\Sigma}\hat{V}^*$:

$$\frac{1}{\sqrt{2}}\underline{\sigma}_sG_2G_2^* - \epsilon I \preceq \hat{V}\hat{\Sigma}\hat{V}^* \preceq \sqrt{2}\overline{\sigma}_sG_2G_2^* + \sqrt{2}\epsilon I \tag{53}$$

---

[3]This is the only place where the diagonalizability of $N_1$ is used in the proof of Theorem 5.3. If $N_1$ is not diagonalizable, $\Lambda$ will be block diagonal consisting of Jordan blocks, and we can still bound $\Lambda^{\frac{m}{2}+j\alpha}(\Lambda^{-\frac{m}{2}-j\alpha})^*$ with dependence on the size of the Jordan block. See Appendix G for more details.

[4]If $\alpha$ does not divide $p$ or $q$, then the last block in (50) will take a slightly different form but can still be bounded similarly.

where we used $\underline{\sigma}_s^2 I \preceq \mathcal{S}_{CB}^* \mathcal{S}_{CB} \preceq \overline{\sigma}_s^2 I$ and $\sigma_{\min}(\mathcal{S}_{CB}) \geq \underline{\sigma}_s$, as $\mathcal{S}_{CB}$ is the block-diagonal matrix of $S_{CB}^{(j)}$ (cf. (49), (50)). Therefore, plugging (53) into (51), we are ready to bound $\hat{S}^* \hat{S}$ has follows:

$$
\begin{aligned}
\hat{S}^* \hat{S} &\preceq \sqrt{2} (P_1^{-1})^* (G_2^* G_2)^{-1} G_2^* (\overline{\sigma}_s G_2 G_2^* + \epsilon I) G_2 (G_2^* G_2)^{-1} P_1^{-1} \\
&= \sqrt{2} \overline{\sigma}_s (P_1^{-1})^* P_1^{-1} + \sqrt{2} \epsilon (\tilde{\mathcal{C}} \tilde{\mathcal{C}}^*)^{-1} \\
&\preceq \sqrt{2} \left( \frac{\overline{\sigma}_s}{\sigma_{\min}(P_1)^2} + \frac{2\epsilon}{\sigma_{\min}(\tilde{\mathcal{C}})^2} \right) I.
\end{aligned}
\tag{54}
$$

Similarly,

$$
\hat{S}^* \hat{S} \succeq \left( \frac{\underline{\sigma}_s}{\sqrt{2}\sigma_{\max}(P_1)^2} - \frac{\epsilon}{\sigma_{\min}(\tilde{\mathcal{C}})^2} \right) I.
\tag{55}
$$

If we further pick $\epsilon$ such that

$$
\epsilon \leq \min\left( \frac{\underline{\sigma}_s \sigma_{\min}(\tilde{\mathcal{C}})^2}{2\sqrt{2}\sigma_{\max}(P_1)^2}, \frac{\overline{\sigma}_s \sigma_{\min}(\tilde{\mathcal{C}})^2}{2\sqrt{2}\sigma_{\min}(P_1)^2} \right) = c_1^2 \sigma_{\min}(\tilde{\mathcal{C}})^2,
\tag{56}
$$

(54) and (55) can be simplified as

$$
\underbrace{\frac{\underline{\sigma}_s}{2\sqrt{2}\sigma_{\max}(P_1)^2}}_{c_1^2} I \preceq \hat{S}^* \hat{S} \preceq \underbrace{2\sqrt{2}\frac{\overline{\sigma}_s}{\sigma_{\min}(P_1)^2}}_{c_2^2} I.
\tag{57}
$$

**Proof of part (c).** Observe that $\tilde{\mathcal{O}}_{1:\alpha} = \mathcal{G}_{\mathrm{ob}} N_1^{m/2}$. Therefore,

$$
\sigma_{\min}(\tilde{\mathcal{O}}) \geq \sigma_{\min}(\tilde{\mathcal{O}}_{1:\alpha}) \geq \sigma_{\min}(\mathcal{G}_{\mathrm{ob}})\sigma_{\min}(N_1^{m/2}) := \tilde{o}_{\min}(m).
$$

Similarly, we have

$$
\sigma_{\min}(\tilde{\mathcal{C}}) \geq \sigma_{\min}(\tilde{\mathcal{C}}_{1:\alpha}) \geq \sigma_{\min}(\mathcal{G}_{\mathrm{con}})\sigma_{\min}(N_1^{m/2}) := \tilde{c}_{\min}(m).
$$

$\square$

We are now ready to prove Lemma B.2

*Proof of Lemma B.2.* Let $E = \tilde{\mathcal{O}} - \hat{\mathcal{O}}\hat{S}$, by Lemma D.1, $\|E\| \leq \frac{\epsilon}{\sigma_{\min}(\tilde{\mathcal{C}})}$. Moreover, $\tilde{\mathcal{O}}_{1:p-1} = \hat{\mathcal{O}}_{1:p-1}\hat{S} + E_{1:p-1}$ and $\tilde{\mathcal{O}}_{2:p} = \hat{\mathcal{O}}_{2:p}\hat{S} + E_{2:p}$. Given that $\tilde{\mathcal{O}}_{2:p} = \tilde{\mathcal{O}}_{1:p-1}N_1$, we have $N_1 = (\tilde{\mathcal{O}}_{1:p-1}^* \tilde{\mathcal{O}}_{1:p-1})^{-1}\tilde{\mathcal{O}}_{1:p-1}^* \tilde{\mathcal{O}}_{2:p}$, where we have used by the condition in the main theorem,

$$
p - 1 \geq \alpha
\tag{58}
$$

so that $\tilde{\mathcal{O}}_{1:p-1}$ is full column rank. Recall $N_1$ is estimated by

$$
\hat{N}_1 := (\hat{\mathcal{O}}_{1:p-1}^* \hat{\mathcal{O}}_{1:p-1})^{-1}\hat{\mathcal{O}}_{1,p-1}^* \hat{\mathcal{O}}_{2:p}.
\tag{59}
$$

Given that $\tilde{\mathcal{O}}_{2:p} = \tilde{\mathcal{O}}_{1:p-1}N_1$, we have

$$
\hat{\mathcal{O}}_{2:p}\hat{S} + E_{2:p} = \left( \hat{\mathcal{O}}_{1:p-1}\hat{S} + E_{1:p-1} \right) N_1
\tag{60}
$$

$$
\Rightarrow \hat{\mathcal{O}}_{2:p} + E_{2:p}\hat{S}^{-1} = \hat{\mathcal{O}}_{1:p-1}\hat{S}N_1\hat{S}^{-1} + E_{1:p-1}N_1\hat{S}^{-1}
\tag{61}
$$

$$
\Rightarrow \hat{\mathcal{O}}_{2:p} = \hat{\mathcal{O}}_{1:p-1}\hat{S}N_1\hat{S}^{-1} + \tilde{E},
\tag{62}
$$

where $\tilde{E} := E_{1:p-1}N_1\hat{S}^{-1} - E_{2:p}\hat{S}^{-1}$. Substituting (62) into (59), we obtain

$$
\begin{aligned}
\hat{N}_1 &= (\hat{\mathcal{O}}_{1:p-1}^* \hat{\mathcal{O}}_{1:p-1})^{-1}\hat{\mathcal{O}}_{1:p-1}^* \left( \hat{\mathcal{O}}_{1:p-1}\hat{S}N_1\hat{S}^{-1} + \tilde{E} \right) \\
&= \hat{S}N_1\hat{S}^{-1} + (\hat{\mathcal{O}}_{1:p-1}^* \hat{\mathcal{O}}_{1:p-1})^{-1}\hat{\mathcal{O}}_{1,p-1}^* \tilde{E}.
\end{aligned}
$$

Therefore,

$$
\begin{aligned}
\left\| N_1 - \hat{S}^{-1}\hat{N}_1\hat{S} \right\| &\leq \left\| \hat{S}^{-1}(\hat{\mathcal{O}}_{1:p-1}^*\hat{\mathcal{O}}_{1:p-1})^{-1}\hat{\mathcal{O}}_{1,p-1}^*\tilde{E}\hat{S} \right\| \\
&= \left\| \hat{S}^{-1}(\hat{\mathcal{O}}_{1:p-1}^*\hat{\mathcal{O}}_{1:p-1})^{-1}\hat{\mathcal{O}}_{1,p-1}^*(E_{1:p-1}N_1 - E_{2:p}) \right\| \\
&\leq \frac{\epsilon}{c_1\sigma_{\min}(\hat{\mathcal{O}}_{1,p-1})}(1+L) \\
&\leq \frac{2L\epsilon}{c_1\sigma_{\min}(\hat{S}^{-1})(\tilde{o}_{\min}(m) - \left\|\tilde{\mathcal{O}} - \hat{\mathcal{O}}\hat{S}\right\|)} \\
&\leq \frac{4Lc_2\epsilon}{c_1\tilde{o}_{\min}(m)}.
\end{aligned}
$$

where the last inequality requires (47). $\qquad\square$

Following similar arguments from the above proof for the estimation of $\widehat{N_{1,O}^{\frac{m}{2}}}$ in (22), we have the following collarary.

**Corollary D.2.** *We have*

$$
\left\| I - \hat{S}^{-1}\widehat{N_{1,O}^{\frac{m}{2}}}\hat{S}N_1^{-\frac{m}{2}} \right\| \leq \frac{4c_2\epsilon}{c_1\tilde{o}_{\min}(m)}.
$$

*Proof.* Let $E = \tilde{\mathcal{O}} - \hat{\mathcal{O}}\hat{S}$, then $\|E\| \leq \frac{\epsilon}{\sigma_{\min}(\tilde{\mathcal{C}})}$. Moreover, $\tilde{\mathcal{O}}_{1:p-\frac{m}{2}} = \hat{\mathcal{O}}_{1:p-\frac{m}{2}}\hat{S} + E_{1:p-\frac{m}{2}}$ and $\tilde{\mathcal{O}}_{\frac{m}{2}+1:p} = \hat{\mathcal{O}}_{\frac{m}{2}+1:p}\hat{S} + E_{\frac{m}{2}+1:p}$. Given that $\tilde{\mathcal{O}}_{\frac{m}{2}+1:p} = \tilde{\mathcal{O}}_{1:p-\frac{m}{2}}N_1^{\frac{m}{2}}$, we have $N_1^{\frac{m}{2}} = (\tilde{\mathcal{O}}_{1:p-\frac{m}{2}}^*\tilde{\mathcal{O}}_{1:p-\frac{m}{2}})^{-1}\tilde{\mathcal{O}}_{1:p-\frac{m}{2}}^*\tilde{\mathcal{O}}_{\frac{m}{2}+1:p}$, where we have used by the condition in the main theorem,

$$
p \geq \alpha + \frac{m}{2} \tag{63}
$$

so that $\tilde{\mathcal{O}}_{1:p-\frac{m}{2}}$ is full column rank. Recall in (22), we estimate $N_1^{\frac{m}{2}}$ by

$$
\widehat{N_{1,O}^{\frac{m}{2}}} := (\hat{\mathcal{O}}_{1:p-\frac{m}{2}}^*\hat{\mathcal{O}}_{1:p-\frac{m}{2}})^{-1}\hat{\mathcal{O}}_{1:p-\frac{m}{2}}^*\hat{\mathcal{O}}_{\frac{m}{2}+1:p}. \tag{64}
$$

Given that $\tilde{\mathcal{O}}_{\frac{m}{2}+1:p} = \tilde{\mathcal{O}}_{1:p-\frac{m}{2}}N_1^{\frac{m}{2}}$, we have

$$
\hat{\mathcal{O}}_{\frac{m}{2}+1:p}\hat{S} + E_{\frac{m}{2}+1:p} = \left( \hat{\mathcal{O}}_{1:p-\frac{m}{2}}\hat{S} + E_{1:p-\frac{m}{2}} \right) N_1^{\frac{m}{2}} \tag{65}
$$

$$
\Rightarrow \hat{\mathcal{O}}_{\frac{m}{2}+1:p} + E_{\frac{m}{2}+1:p}\hat{S}^{-1} = \hat{\mathcal{O}}_{1:p-\frac{m}{2}}\hat{S}N_1^{\frac{m}{2}}\hat{S}^{-1} + E_{1:p-\frac{m}{2}}N_1^{\frac{m}{2}}\hat{S}^{-1} \tag{66}
$$

$$
\Rightarrow \hat{\mathcal{O}}_{\frac{m}{2}+1:p} = \hat{\mathcal{O}}_{1:p-\frac{m}{2}}\hat{S}N_1^{\frac{m}{2}}\hat{S}^{-1} + \tilde{E}, \tag{67}
$$

where $\tilde{E} := E_{1:p-\frac{m}{2}}N_1^{\frac{m}{2}}\hat{S}^{-1} - E_{\frac{m}{2}+1:p}\hat{S}^{-1}$. Substituting (67) into (64), we obtain

$$
\widehat{N_{1,O}^{\frac{m}{2}}} = (\hat{\mathcal{O}}_{1:p-\frac{m}{2}}^*\hat{\mathcal{O}}_{1:p-\frac{m}{2}})^{-1}\hat{\mathcal{O}}_{1:p-\frac{m}{2}}^*(\hat{\mathcal{O}}_{1:p-\frac{m}{2}}\hat{S}N_1^{\frac{m}{2}}\hat{S}^{-1} + \tilde{E}) \tag{68}
$$

$$
= \hat{S}N_1^{\frac{m}{2}}\hat{S}^{-1} + (\hat{\mathcal{O}}_{1:p-\frac{m}{2}}^*\hat{\mathcal{O}}_{1:p-\frac{m}{2}})^{-1}\hat{\mathcal{O}}_{1:p-\frac{m}{2}}^*\tilde{E}. \tag{69}
$$

Therefore,

$$
\begin{aligned}
\left\| I - \hat{S}^{-1}\widehat{N_{1,O}^{\frac{m}{2}}}\hat{S}N_1^{-\frac{m}{2}} \right\| &= \left\| \hat{S}^{-1}(\hat{\mathcal{O}}_{1:p-\frac{m}{2}}^*\hat{\mathcal{O}}_{1:p-\frac{m}{2}})^{-1}\hat{\mathcal{O}}_{1:p-\frac{m}{2}}^*\tilde{E}\hat{S}N_1^{-\frac{m}{2}} \right\| \\
&= \left\| \hat{S}^{-1}(\hat{\mathcal{O}}_{1:p-\frac{m}{2}}^*\hat{\mathcal{O}}_{1:p-\frac{m}{2}})^{-1}\hat{\mathcal{O}}_{1:p-\frac{m}{2}}^*(E_{1:p-\frac{m}{2}}N_1^{\frac{m}{2}}\hat{S}^{-1} - E_{\frac{m}{2}+1:p}\hat{S}^{-1})\hat{S}N_1^{-\frac{m}{2}} \right\| \\
&\leq \frac{\epsilon}{c_1\sigma_{\min}(\hat{\mathcal{O}}_{1:p-\frac{m}{2}})}\left( 1 + L(\frac{\frac{1}{|\lambda_k|}+1}{2})^{\frac{m}{2}} \right) \\
&\leq \frac{2\epsilon}{c_1\sigma_{\min}(\hat{\mathcal{O}}_{1:p-\frac{m}{2}})} \tag{70}
\end{aligned}
$$

$$\leq \frac{2\epsilon}{c_1\sigma_{\min}(\hat{S}^{-1})(\tilde{o}_{\min}(m) - \left\|\tilde{\mathcal{O}} - \hat{\mathcal{O}}\hat{S}\right\|)}$$

$$\leq \frac{2\epsilon}{c_1\sigma_{\min}(\hat{S}^{-1})(\tilde{o}_{\min}(m) - \frac{1}{\sigma_{\min}(\tilde{\mathcal{C}})}\epsilon)}$$

$$\leq \frac{4c_2\epsilon}{c_1\tilde{o}_{\min}(m)}, \tag{71}$$

where (70) requires

$$m > \frac{-\log(L)}{\log(\frac{\frac{1}{|\lambda_k|}+1}{2})}, \tag{72}$$

which we will verify at the end of the proof of Theorem 5.3, and (71) requires (47). $\qquad\square$

By using an idental argument, we can also prove a similar corollary for $\widehat{N_{1,C}^{\frac{m}{2}}}$. The proof is omitted.

**Corollary D.3.** *Assuming $p, q > \frac{m}{2} + 1$, we have*

$$\left\|I - N_1^{-\frac{m}{2}}\hat{S}^{-1}\widehat{N_{1,C}^{\frac{m}{2}}}\hat{S}\right\| \leq \frac{4c_2\epsilon}{c_1\tilde{c}_{\min}(m)}.$$

With the above two corallaries, we are now ready to prove Lemma B.3.

*Proof of Lemma B.3.* By the definition of $\tilde{\mathcal{O}}$ in (7), we have

$$\tilde{\mathcal{O}}_1 = CQ_1 N_1^{\frac{m}{2}}. \tag{73}$$

Furthermore, we recall by the way $\hat{C}$ is estimated in (24), we obtain

$$\hat{\mathcal{O}}_1 = \hat{C}\widehat{N_{1,O}^{\frac{m}{2}}}. \tag{74}$$

Let $E_{O_1} := \tilde{\mathcal{O}}_1 - \hat{\mathcal{O}}_1\hat{S}$. Substituting it in (74) leads to

$$\tilde{\mathcal{O}}_1 = \hat{\mathcal{O}}_1\hat{S} + E_{O_1} = \hat{C}\widehat{N_{1,O}^{\frac{m}{2}}}\hat{S} + E_{O_1}.$$

Substituting in (73), we have

$$CQ_1 N_1^{\frac{m}{2}} = \hat{C}\widehat{N_{1,O}^{\frac{m}{2}}}\hat{S} + E_{O_1}$$

$$\Rightarrow CQ_1 = \hat{C}\hat{S}\hat{S}^{-1}\widehat{N_{1,O}^{\frac{m}{2}}}\hat{S}N_1^{-\frac{m}{2}} + E_{O_1}N_1^{-\frac{m}{2}}. \tag{75}$$

By Corollary D.2, we obtain

$$\left\|CQ_1 - \hat{C}\hat{S}\right\| \leq \left\|\hat{C}\hat{S}\right\|\left\|\hat{S}^{-1}\widehat{N_{1,O}^{\frac{m}{2}}}\hat{S}N_1^{-m/2} - I\right\| + \|E_{O_1}\|\left\|N_1^{-m/2}\right\|$$

$$\leq \left\|\hat{C}\hat{S}\right\|\frac{4c_2\epsilon}{c_1\tilde{o}_{\min}(m)} + \|E_{O_1}\|\left\|N_1^{-m/2}\right\|$$

$$\leq \left(\left\|\hat{C}\hat{S}\right\|\frac{4c_2}{c_1\tilde{o}_{\min}(m)} + \frac{1}{\tilde{c}_{\min}(m)}\right)\epsilon \tag{76}$$

$$\leq \left(\frac{4c_2(\left\|CQ_1 - \hat{C}\hat{S}\right\| + \|CQ_1\|)}{c_1\tilde{o}_{\min}(m)} + \frac{1}{\tilde{c}_{\min}(m)}\right)\epsilon \tag{77}$$

$$\leq 2\left(\frac{4c_2 L}{c_1\tilde{o}_{\min}(m)} + \frac{1}{\tilde{c}_{\min}(m)}\right)\epsilon. \tag{78}$$

where (76) is by Lemma D.1 and requires (72), and (78) requires

$$\epsilon < \frac{c_1\tilde{o}_{\min}(m)}{8c_2}. \tag{79}$$

$\qquad\square$

The proof of Lemma B.4 undergoes a very similar procedure.

*Proof of Lemma B.4.* By the definition of $\tilde{\mathcal{C}}$ in (7), we have

$$\tilde{\mathcal{C}}_1 = N_1^{\frac{m}{2}} R_1 B. \tag{80}$$

Further, by (24), we have

$$\hat{\mathcal{C}}_1 = \widehat{N_{1,C}^{\frac{m}{2}}} \hat{B}. \tag{81}$$

Let $E_{C_1} := \tilde{\mathcal{C}}_1 - \hat{S}^{-1} \hat{\mathcal{C}}_1$. Substitute in (81), we get

$$\tilde{\mathcal{C}}_1 = \hat{S}^{-1} \hat{\mathcal{C}}_1 + E_{C_1} = \hat{S}^{-1} \widehat{N_{1,C}^{\frac{m}{2}}} \hat{B} + E_{C_1}.$$

Substitute in (80), we obtain

$$N_1^{\frac{m}{2}} R_1 B = \hat{S}^{-1} \widehat{N_{1,C}^{\frac{m}{2}}} \hat{B} + E_{C_1}$$

$$R_1 B = N_1^{-\frac{m}{2}} \hat{S}^{-1} \widehat{N_{1,C}^{\frac{m}{2}}} \hat{S} \hat{S}^{-1} \hat{B} + N_1^{-\frac{m}{2}} E_{C_1}. \tag{82}$$

By Corollary D.3, we obtain

$$\left\| R_1 B - \hat{S}^{-1} \hat{B} \right\| \leq \left\| N_1^{-\frac{m}{2}} \hat{S}^{-1} \widehat{N_{1,C}^{\frac{m}{2}}} \hat{S} - I \right\| \left\| \hat{S}^{-1} \hat{B} \right\| + \left\| N_1^{-\frac{m}{2}} \right\| \left\| E_{C_1} \right\|$$

$$\leq 4 \left( \frac{c_2 \left\| \hat{S}^{-1} \hat{B} \right\|}{c_1 \tilde{c}_{\min}(m)} + \frac{1}{\tilde{o}_{\min}(m)} \right) \epsilon \tag{83}$$

$$\leq 4 \left( \frac{c_2 (\left\| R_1 B - \hat{S}^{-1} \hat{B} \right\| + \|R_1 B\|)}{c_1 \tilde{c}_{\min}(m)} + \frac{1}{\tilde{o}_{\min}(m)} \right) \epsilon \tag{84}$$

$$\leq 8 \left( \frac{c_2 L}{c_1 \tilde{c}_{\min}(m)} + \frac{1}{\tilde{o}_{\min}(m)} \right) \epsilon, \tag{85}$$

where (83) is by Lemma D.1 and requires (72), and (85) requires

$$\epsilon < \frac{c_1 \tilde{c}_{\min}(m)}{8c_2}. \tag{86}$$

$\square$

# E   Proof of Step 3

Before we analyze the analytic condition of the transfer functions, we need to analyze the values of $F(z)$ and $\hat{F}(z)$ on the unit circle through the following lemma:

**Lemma E.1.** *We have*

$$\sup_{|z|=1} \left\| F(z) - \hat{F}(z) \right\| \leq \frac{1}{|\lambda_k| - 1} \left( 8L^2 \epsilon_C + \frac{4L^4 \epsilon_N}{|\lambda_k| - 1} + L^2 \epsilon_B \right).$$

*Proof.* We have that

$$\sup_{|z|=1} \left\| \hat{F}(z) - F(z) \right\| = \sup_{|z|=1} \left\| \hat{C} \hat{S} (zI - \hat{S}^{-1} \hat{N}_1 \hat{S})^{-1} \hat{S}^{-1} \hat{B} - CQ_1 (zI - N_1)^{-1} R_1 B \right\|$$

$$\leq \sup_{|z|=1} \Big( \left\| \hat{C} \hat{S} (zI - \hat{S}^{-1} \hat{N}_1 \hat{S})^{-1} \hat{S}^{-1} \hat{B} - CQ_1 (zI - \hat{S}^{-1} \hat{N}_1 \hat{S})^{-1} \hat{S}^{-1} \hat{B} \right\|$$

$$+ \left\| CQ_1 (zI - \hat{S}^{-1} \hat{N}_1 \hat{S})^{-1} \hat{S}^{-1} \hat{B} - CQ_1 (zI - N_1)^{-1} \hat{S}^{-1} \hat{B} \right\| + \left\| CQ_1 (zI - N_1)^{-1} \hat{S}^{-1} \hat{B} - CQ_1 (zI - N_1)^{-1} R_1 B \right\| \Big)$$

$$\leq \sup_{|z|=1} \Big( \left\| \hat{C} \hat{S} - CQ_1 \right\| \left\| (zI - \hat{S}^{-1} \hat{N}_1 \hat{S})^{-1} \hat{S}^{-1} \hat{B} \right\| + \|CQ_1\| \left\| (zI - \hat{S}^{-1} \hat{N}_1 \hat{S})^{-1} - (zI - N_1)^{-1} \right\| \left\| \hat{S}^{-1} \hat{B} \right\|$$

$$+ \left\| CQ_1(zI - N_1)^{-1} \right\| \left\| \hat{S}^{-1}\hat{B} - R_1 B \right\| \Big). \tag{87}$$

Moreover,

$$\left\| (zI - \hat{S}^{-1}\hat{N}_1\hat{S})^{-1} - (zI - N_1)^{-1} \right\| \le \left\| (zI - \hat{S}^{-1}\hat{N}_1\hat{S})^{-1} \right\| \left\| \hat{S}^{-1}\hat{N}_1\hat{S} - N_1 \right\| \left\| (zI - N_1)^{-1} \right\|. \tag{88}$$

Substituting (88) into (87), we obtain

$$
\begin{aligned}
\left\| \hat{F}(z) - F(z) \right\|_{H_\infty} &\le \sup_{|z|=1} \Big( \epsilon_C \left\| (zI - \hat{S}^{-1}\hat{N}_1\hat{S})^{-1}\hat{S}^{-1}\hat{B} \right\| + \epsilon_N \left\| CQ_1 \right\| \left\| (zI - \hat{S}^{-1}\hat{N}_1\hat{S})^{-1} \right\| \left\| (zI - N_1)^{-1} \right\| \left\| \hat{S}^{-1}\hat{B} \right\| \\
&\qquad + \epsilon_B \left\| CQ_1(zI - N_1)^{-1} \right\| \Big) \\
&\le \sup_{|z|=1} \Big( 4 \left\| \hat{S}^{-1}\hat{B} \right\| \left\| (zI - N_1)^{-1} \right\| \epsilon_C + 2 \left\| CQ_1 \right\| \left\| \hat{S}^{-1}\hat{B} \right\| \left\| (zI - N_1)^{-1} \right\|^2 \epsilon_N + \left\| CQ_1 \right\| \left\| (zI - N_1)^{-1} \right\| \epsilon_B \Big) \tag{89} \\
&\le \sup_{|z|=1} \Big( 8L \left\| (zI - N_1)^{-1} \right\| \epsilon_C + 4 \left\| CQ_1 \right\| L \left\| (zI - N_1)^{-1} \right\|^2 \epsilon_N + \left\| CQ_1 \right\| \left\| (zI - N_1)^{-1} \right\| \epsilon_B \Big) \tag{90}
\end{aligned}
$$

where (89) requires $\left\| (zI - N_1 + N_1 - \hat{S}^{-1}\hat{N}_1\hat{S})^{-1} \right\| = \frac{1}{\sigma_{\min}(zI - N_1 + N_1 - \hat{S}^{-1}\hat{N}_1\hat{S})} \le \frac{1}{\sigma_{\min}(zI - N_1) - \epsilon_N} \le \frac{2}{\sigma_{\min}(zI - N_1)} = 2 \left\| (zI - N_1)^{-1} \right\|$, which in turn requires, by Lemma B.2,

$$\epsilon_N \le \frac{1}{2} \inf_{|z|=1} \sigma_{\min}(zI - N_1) \quad \Leftarrow \quad \epsilon \le \frac{c_1 \tilde{o}_{\min}(m)}{8Lc_2} \inf_{|z|=1} \sigma_{\min}(zI - N_1) \Leftarrow \epsilon \le \frac{c_1 \tilde{o}_{\min}(m)}{8L^2 c_2}(|\lambda_k| - 1), \tag{91}$$

where the last step uses $\inf_{|z|=1} \sigma_{\min}(zI - N_1) = \frac{1}{\sup_{|z|=1} \|(zI - N_1)^{-1}\|} \ge \frac{|\lambda_k| - 1}{L}$. Eq. (90) uses $\left\| \hat{S}^{-1}\hat{B} \right\| \le \|R_1 B\| + \epsilon_B \le 2L$, which requires

$$\epsilon_B \le L \quad \Leftarrow \quad \epsilon \le \min\Big(\frac{L}{16}\frac{c_1 \tilde{c}_{\min}(m)}{c_2 L}, \frac{L}{16}\tilde{o}_{\min}(m)\Big). \tag{92}$$

We will verify (91) and (92) at the end of the proof of Theorem 5.3 to make sure it is met by our choice of algorithm parameters. Lastly, substituting $\max(\|R_1 B\|, \|CQ_1\|) \le L$ and using $\sup_{|z|=1} \left\| (zI - N_1)^{-1} \right\| \le \frac{L}{|\lambda_k| - 1}$ give the desired result. $\qquad\square$

We are now ready to prove Proposition B.5,

*Proof of Proposition B.5.* We only prove part (a). The proof of part (b) is similar and is hence omitted. Suppose $K(z)$ is a controller that satisfies the condition in part(a). Our goal is to show this $K(z)$ stabilizes $\hat{F}(z)$, or in other words, $\hat{F}_K(z)$ is analytic on $\mathbb{C}_{\ge 1}$. Let $\Delta_F(z) := F(z) - \hat{F}(z)$, we have

$$
\begin{aligned}
I - \hat{F}(z)K(z) &= I - F(z)K(z) + \Delta_F(z)K(z) \\
&= (I + \Delta_F(z)K(z)(I - F(z)K(z))^{-1})(I - F(z)K(z)).
\end{aligned}
$$

Note that to show that $(I - \hat{F}(z)K(z))^{-1}$ is analytic on $\mathbb{C}_{\ge 1}$, we only need to show $(I - \hat{F}(z)K(z))$ has the same number of zeros on $\mathbb{C}_{\ge 1}$ as that of $I - F(z)K(z)$, because we know $(I - F(z)K(z))$ is non-singular (i.e. has no zeros) on $\mathbb{C}_{\ge 1}$. Given (91), we also know $I - F(z)K(z)$ and $I - \hat{F}(z)K(z)$ has the same number of poles on $\mathbb{C}_{\ge 1}$. By Rouche's Theorem (Theorem H.4), $I - \hat{F}(z)K(z)$ has the same number of zeros on $\mathbb{C}_{\ge 1}$ as $I - F(z)K(z)$ if

$$
\begin{aligned}
&\|(I - \hat{F}(z)K(z)) - (I - F(z)K(z))\| < \|(I - F(z)K(z))\|, \forall |z| = 1 \\
&\Leftarrow \sup_{|z|=1} \left\| \Delta_F(z)K(z)(I - F(z)K(z))^{-1} \right\| < 1 \\
&\Leftarrow \sup_{|z|=1} \|\Delta_F(z)\| \left\| K(z)(I - F(z)K(z))^{-1} \right\| \le \epsilon_* \sup_{|z|=1} \|K(z)F_K(z)\| < 1,
\end{aligned}
$$

where the last condition is satisfied by the condition in part (a) of this proposition. Therefore, $I - \hat{F}(z)K(z)$ has no zeros on $\mathbb{C}_{\ge 1}$ and the proof is concluded.

$$\square$$

We are now ready to prove Theorem 5.3.

*Proof of Theorem 5.3.* Suppose for now the number of samples is large enough such that the condition in Proposition B.5 holds: $\sup_{|z|=1} \left\| F(z) - \hat{F}(z) \right\| < \epsilon_*$. At the end of the proof we will provide a sample complexity bound for which this holds.

By Assumption 5.2, there exists $K(z)$ that stabilizes $F(z)$ with

$$\|K(z)F_K(z)\|_{H_\infty} \leq \frac{1}{\|\Delta(z)\|_{H_\infty} + 3\epsilon_*}.$$

Therefore, by Proposition B.5, $K(z)$ stabilizes plant $\hat{F}(z)$ as well. Therefore, $\|K(z)\hat{F}_K(z)\|_{H_\infty}$ is well defined and can be evaluated by only taking sup over $|z| = 1$:

$$\|K(z)\hat{F}_K(z)\|_{H_\infty} = \left\| K(z)(I - \hat{F}(z)K(z))^{-1} \right\|_{H_\infty}$$

$$= \sup_{|z|=1} \left\| K(z)(I - \hat{F}(z)K(z))^{-1} \right\|_2$$

$$= \sup_{|z|=1} \left\| K(z)(I - F(z)K(z) + F(z)K(z) - \hat{F}(z)K(z))^{-1} \right\|_2$$

$$= \sup_{|z|=1} \left\| K(z)(I - F(z)K(z) + \Delta_F(z)K(z))^{-1} \right\|_2$$

$$= \sup_{|z|=1} \left\| K(z)\underbrace{(I - F(z)K(z))^{-1}}_{F_K(z)}(I + \Delta_F(z)K(z)\underbrace{(I - F(z)K(z))^{-1}}_{F_K(z)})^{-1} \right\|_2$$

$$= \sup_{|z|=1} \left\| K(z)F_K(z)(I + \Delta_F(z)K(z)F_K(z))^{-1} \right\|_2$$

$$\leq \sup_{|z|=1} \|K(z)F_K(z)\| \sum_{i=0}^{\infty} \|\Delta_F(z)\|^i \|K(z)F_K(z)\|^i$$

$$= \sup_{|z|=1} \frac{\|K(z)F_K(z)\|}{1 - \|\Delta_F(z)\| \|K(z)F_K(z)\|}$$

$$= \frac{\|K(z)F_K(z)\|_{H_\infty}}{1 - \epsilon_*\|K(z)F_K(z)\|_{H_\infty}}$$

$$\leq \frac{1}{\|\Delta(z)\|_{H_\infty} + 2\epsilon_*}, \tag{93}$$

where we substituted in Assumption 5.2 in the last inequality. In Algorithm 1, we find a controller $\hat{K}(z)$ that stabilizes $\hat{F}(z)$ and minimizes $\|\hat{K}(z)\hat{F}_{\hat{K}}(z)\|_{H_\infty}$. As $K(z)$ stabilizes $\hat{F}(z)$ and satisfies the upper bound in (93), the $\hat{K}(z)$ returned by the algorithm must stabilize $\hat{F}(z)$ and is such that $\|\hat{K}(z)\hat{F}_{\hat{K}}(z)\|_{H_\infty}$ is upper bounded by the RHS of (93). By Proposition B.5, we know $\hat{K}(z)$ stabilizes $F(z)$ as well. Therefore, $\|\hat{K}(z)F_{\hat{K}}(z)\|_{H_\infty}$ is well defined and can be evaluated on taking the sup over $|z| = 1$:

$$\|\hat{K}(z)F_{\hat{K}}(z)\|_{H_\infty} = \left\| \hat{K}(z)(I - F(z)\hat{K}(z))^{-1} \right\|_{H_\infty}$$

$$= \sup_{|z|=1} \left\| \hat{K}(z)(I - F(z)\hat{K}(z))^{-1} \right\|_2$$

$$= \sup_{|z|=1} \left\| \hat{K}(z)(I - \hat{F}(z)\hat{K}(z) + \hat{F}(z)\hat{K}(z) - F(z)\hat{K}(z))^{-1} \right\|_2$$

$$= \sup_{|z|=1} \left\| \hat{K}(z)(I - \hat{F}(z)\hat{K}(z) - \Delta_F(z)\hat{K}(z))^{-1} \right\|_2$$

$$= \sup_{|z|=1} \left\| \hat{K}(z) \underbrace{(I - \hat{F}(z)\hat{K}(z))^{-1}}_{\hat{F}_{\hat{K}}(z)} (I - \Delta_F(z)\hat{K}(z) \underbrace{(I - \hat{F}(z)\hat{K}(z))^{-1}}_{\hat{F}_{\hat{K}}(z)})^{-1} \right\|_2$$

$$= \sup_{|z|=1} \left\| \hat{K}(z)\hat{F}_{\hat{K}}(z)(I - \Delta_F(z)\hat{K}(z)\hat{F}_{\hat{K}}(z))^{-1} \right\|_2$$

$$\leq \left\| \hat{K}(z)\hat{F}_{\hat{K}}(z) \right\|_{H_\infty} \sup_{|z|=1} \sum_{i=0}^{\infty} \|\Delta_F(z)\|^i \left\| \hat{K}(z)\hat{F}_{\hat{K}}(z) \right\|^i$$

$$= \frac{\left\| \hat{K}(z)\hat{F}_{\hat{K}}(z) \right\|_{H_\infty}}{1 - \epsilon_* \left\| \hat{K}(z)\hat{F}_{\hat{K}}(z) \right\|_{H_\infty}}. \tag{94}$$

As $\hat{K}(z)$ is such that $\|\hat{K}(z)\hat{F}_{\hat{K}}(z)\|_{H_\infty}$ satisfies the upper bound by the RHS in (93), we have

$$\frac{\left\| \hat{K}(z)\hat{F}_{\hat{K}}(z) \right\|_{H_\infty}}{1 - \epsilon_* \left\| \hat{K}(z) \right\| \hat{F}_K(z)}_{H_\infty} \leq \frac{\frac{1}{\|\Delta(z)\|_{H_\infty} + 2\epsilon_*}}{1 - \epsilon_* \frac{1}{\|\Delta(z)\|_{H_\infty} + 2\epsilon_*}} \leq \frac{1}{\|\Delta(z)\|_{H_\infty} + \epsilon_*} < \frac{1}{\|\Delta(z)\|_{H_\infty}}. \tag{95}$$

Therefore, by small gain theorem (Lemma 3.1), we know the controller $\hat{K}(z)$ stabilizes the original full system.

For the remaining of the proof, we analyze the number of samples required to satisfy the condition in Proposition B.5, i.e.

$$\sup_{|z|=1} \left\| F(z) - \hat{F}(z) \right\| < \epsilon_*$$

$$\Leftarrow \frac{1}{|\lambda_k| - 1}\left( 8L^2\epsilon_C + \frac{4L^4\epsilon_N}{|\lambda_k| - 1} + L^2\epsilon_B \right) < \epsilon_*$$

$$\Leftarrow \begin{cases} \epsilon_N & < \frac{(|\lambda_k| - 1)^2\epsilon_*}{12L^4} \\ \epsilon_B & < \frac{(|\lambda_k| - 1)\epsilon_*}{3L^2} \\ \epsilon_C & < \frac{(|\lambda_k| - 1)\epsilon_*}{24L^2} \end{cases}$$

where the middle step is by Lemma E.1. By Lemma B.2, Lemma B.4 and Lemma B.3, we obtain the following sufficient condition,

$$\frac{4Lc_2}{c_1\tilde{o}_{\min}(m)}\epsilon \leq \frac{(|\lambda_k| - 1)^2\epsilon_*}{12L^4},$$

$$8\left( \frac{c_2L}{c_1\tilde{c}_{\min}(m)} + \frac{1}{\tilde{o}_{\min}(m)} \right)\epsilon < \frac{(|\lambda_k| - 1)\epsilon_*}{3L^2},$$

$$2\left( \frac{4c_2L}{c_1\tilde{o}_{\min}(m)} + \frac{1}{\tilde{c}_{\min}(m)} \right)\epsilon < \frac{(|\lambda_k| - 1)\epsilon_*}{24L^2}.$$

By simple computation, we obtain the following sufficient condition:

$$\epsilon < \min\left\{ \frac{c_1\tilde{o}_{\min}(m)(|\lambda_k| - 1)^2\epsilon_*}{48c_2L^5}, \frac{c_1\tilde{c}_{\min}(m)(|\lambda_k| - 1)\epsilon_*}{48c_2L^3}, \frac{\tilde{o}_{\min}(m)(|\lambda_k| - 1)\epsilon_*}{48L^2}, \frac{c_1\tilde{o}_{\min}(m)(|\lambda_k| - 1)\epsilon_*}{384c_2L^3}, \right.$$

$$\left. \frac{\tilde{c}_{\min}(m)(|\lambda_k| - 1)\epsilon_*}{96L^3} \right\},$$

which can be simplified into the following sufficient condition:

$$\epsilon < \frac{c_1}{384c_2L^5} \min(\tilde{o}_{\min}(m), \tilde{c}_{\min}(m)) \min((|\lambda_k| - 1)^2, |\lambda_k| - 1)\epsilon_* \tag{96}$$

Moreover, we also need to satisfy (47),(56),(79),(86),(91),(92), which requires

$$\epsilon < \left\{ \frac{1}{4}\tilde{o}_{\min}(m)\tilde{c}_{\min}(m), c_1^2\tilde{c}_{\min}(m)^2, \frac{c_1\tilde{o}_{\min}(m)}{8c_2}, \frac{c_1\tilde{c}_{\min}(m)}{8c_2}, \right.$$

$$\left. \frac{c_1\tilde{o}_{\min}(m)}{8L^2c_2}(|\lambda_k| - 1), \frac{L}{16}\frac{c_1\tilde{c}_{\min}(m)}{c_2L}, \frac{L}{16}\tilde{o}_{\min}(m) \right\}. \tag{97}$$

A sufficient condition that merges both (96) and (97) is:

$$\epsilon < \frac{c_1}{384 c_2 L^5} \min(\tilde{o}_{\min}(m), \tilde{c}_{\min}(m)) \min((|\lambda_k| - 1)^2, |\lambda_k| - 1) \min(\epsilon_*, 1). \qquad (98)$$

Plugging in the definition of $c_1, c_2, \tilde{o}_{\min}(m), \tilde{c}_{\min}(m)$, we have $\frac{c_1}{c_2} = \sqrt{\frac{\sigma_{\min}(P_1)^2 \sigma_s}{8 \sigma_{\max}(P_1)^2 \bar{\sigma}_s}} = \sqrt{\frac{\sigma_{\min}(P_1)^4 \sigma_{\min}(\mathcal{G}_{\text{ob}}) \sigma_{\min}(\mathcal{G}_{\text{con}})}{8 \sigma_{\max}(P_1)^4 \sigma_{\max}(\mathcal{G}_{\text{ob}}) \sigma_{\max}(\mathcal{G}_{\text{con}})}} = \frac{1}{\kappa(P_1)^2 \sqrt{8 \kappa(\mathcal{G}_{\text{ob}}) \kappa(\mathcal{G}_{\text{con}})}}$, where we recall $\kappa(\cdot)$ means the condition number of a matrix. Further, as $\|(N_1^{-1})^{m/2}\| \leq L(\frac{\frac{1}{|\lambda_k|} + 1}{2})^{m/2}$, we have $\sigma_{\min}(N_1^{m/2}) \geq \frac{1}{L}(\frac{2}{\frac{1}{|\lambda_k|} + 1})^{m/2}$. Therefore, the condition (98) can be replaced with the following sufficient condition

$$\epsilon \lesssim \frac{1}{\sqrt{\kappa(\mathcal{G}_{\text{ob}}) \kappa(\mathcal{G}_{\text{con}})} L^8} \min(\sigma_{\min}(\mathcal{G}_{\text{ob}}), \sigma_{\min}(\mathcal{G}_{\text{con}}))(\frac{2|\lambda_k|}{|\lambda_k| + 1})^{m/2} \min((|\lambda_k| - 1)^2, |\lambda_k| - 1) \min(\epsilon_*, 1).$$
$$(99)$$

where the $\lesssim$ above only hides numerical constants.

To meet (58), (63), we set $p = q = \max(\alpha + \frac{m}{2}, m)$. We also set $m$ large enough so that the right hand side of (99) is lower bounded by $\sqrt{m^3}$. Using the simple fact that for $a > 1$, $a^{m/2} = \sqrt{a}^{m/2} \sqrt{a}^{m/2} \geq \sqrt{a}^{m/2} m^{1.5} e^{1.5} (\frac{1}{6} \log a)^{1.5}$, and replacing $a$ with $\frac{2|\lambda_k|}{|\lambda_k| + 1}$, a sufficient condition for such an $m$ is

$$m \gtrsim \frac{1}{\log \frac{2|\lambda_k|}{|\lambda_k| + 1}} \log \frac{\sqrt{\kappa(\mathcal{G}_{\text{ob}}) \kappa(\mathcal{G}_{\text{con}})} L^8}{\min(\sigma_{\min}(\mathcal{G}_{\text{ob}}), \sigma_{\min}(\mathcal{G}_{\text{con}}))(\log \frac{2|\lambda_k|}{|\lambda_k| + 1})^{3/2} \min((|\lambda_k| - 1)^2, |\lambda_k| - 1) \min(\epsilon_*, 1)},$$
$$(100)$$

Recall that $\epsilon = 2\hat{\epsilon} + 2\tilde{\epsilon}$. With the above condition on $m$, it now suffices to require $\max(\hat{\epsilon}, \tilde{\epsilon}) \lesssim \sqrt{m^3}$. Plugging in the definition of $\hat{\epsilon}, \tilde{\epsilon}$ in Theorem C.2, Lemma C.3, we need

$$M > 8 d_u T + 4(d_u + d_y + 4) \log(3T/\delta) \qquad (101)$$

$$\frac{8 \sigma_v \sqrt{T(T + 1)(d_u + d_y) \min\{p, q\} \log(27T/\delta)}}{\sigma_u \sqrt{M}} \lesssim \sqrt{m^3} \qquad (102)$$

$$L^3 (\frac{|\lambda_{k+1}| + 1}{2})^m \lesssim \sqrt{m^3}. \qquad (103)$$

To satisfy (102), recall $T = m + p + q + 2$. Let us also require

$$m \geq 2\alpha \qquad (104)$$

so that $p = q = \max(\alpha + \frac{m}{2}, m) = m$, so we have $T(T + 1) \min(p, q) \lesssim m^3$. Therefore, it suffices to require $M \gtrsim (\frac{\sigma_v}{\sigma_u})^2 (d_u + d_y) \log \frac{m}{\delta}$ to satisfy (102).

To satisfy (103), we need

$$m \gtrsim \frac{\log \frac{1}{L}}{\log \frac{|\lambda_{k+1}| + 1}{2}}. \qquad (105)$$

To summarize, collecting the requirements on $m$ (100),(104),(105) and also (72), we have the final complexity on $m$:

$$m \gtrsim \max(\alpha, \frac{\log \frac{1}{L}}{\log \frac{|\lambda_{k+1}| + 1}{2}}, \frac{\log(\frac{1}{L})}{\log(\frac{\frac{1}{|\lambda_k|} + 1}{2})}, \qquad (106)$$

$$\frac{1}{\log \frac{2|\lambda_k|}{|\lambda_k| + 1}} \log \frac{\sqrt{\kappa(\mathcal{G}_{\text{ob}}) \kappa(\mathcal{G}_{\text{con}})} L^8}{\min(\sigma_{\min}(\mathcal{G}_{\text{ob}}), \sigma_{\min}(\mathcal{G}_{\text{con}}))(\log \frac{2|\lambda_k|}{|\lambda_k| + 1})^{3/2} \min((|\lambda_k| - 1)^2, |\lambda_k| - 1) \min(\epsilon_*, 1)}).$$
$$(107)$$

The final requirement on $p, q$ is,

$$p = q = m. \qquad (108)$$

The final complexity on the number of trajectories is

$$M \gtrsim (\frac{\sigma_v}{\sigma_u})^2 (d_u + d_y) \log \frac{m}{\delta} + d_u m \tag{109}$$

Lastly, in the most interesting regime that $|\lambda_k| - 1$, $1 - |\lambda_{k+1}|$, $\epsilon_*$ is small, we have $\log \frac{2|\lambda_k|}{|\lambda_k|+1} \asymp |\lambda_k| - 1$, $\min((|\lambda_k| - 1)^2, |\lambda_k| - 1) = (|\lambda_k| - 1)^2$, $\min(\epsilon_*, 1) = \epsilon_*$. In this case, (106) can be simplified:

$$m \gtrsim \max \left( \alpha, \frac{\log L}{1 - |\lambda_{k+1}|}, \frac{1}{|\lambda_k| - 1} \log \frac{\kappa(\mathcal{G}_{\text{ob}})\kappa(\mathcal{G}_{\text{con}})L}{\min(\sigma_{\min}(\mathcal{G}_{\text{ob}}), \sigma_{\min}(\mathcal{G}_{\text{con}}))(|\lambda_k| - 1)\epsilon_*} \right). \tag{110}$$

$\square$

# F    System identification when $D \neq 0$

In the case when $D \neq 0$ in (1), the transfer function becomes

$$F(z) = D + CQ_1(zI - N_1)^{-1}R_1 B, \tag{111}$$

and the estimated transfer function becomes $\hat{F}(z) = \hat{D} + \hat{C}\hat{S}(zI - \hat{S}^{-1}\hat{N}_1\hat{S})^{-1}\hat{S}^{-1}\hat{B}$, so the gap between estimated transfer function and the ground-truth transfer function can be bounded as follows

$$\left\| F(z) - \hat{F}(z) \right\| \leq \left\| D - \hat{D} \right\| + \left\| CQ_1(zI - N_1)^{-1}R_1 B - \hat{C}\hat{S}(zI - \hat{S}^{-1}\hat{N}_1\hat{S})^{-1}\hat{S}^{-1}\hat{B} \right\|. \tag{112}$$

The second term in (112) can be bounded by Lemma E.1. Therefore, in this section, we focus on providing a bound for $\left\| D - \hat{D} \right\|$. The recursive relationship of the process and measurement data will change from (14) to the following:

$$\begin{aligned} y_t^{(i)} &= Cx_t^{(i)} + Du_t^{(i)} + v_t^{(i)} \\ &= \sum_{j=0}^{T} CA^j \left( Bu_{t-j-1}^{(i)} + w_{t-j-1}^{(i)} \right) + Du_t^{(i)} + v_t^{(i)}. \end{aligned} \tag{113}$$

Therefore, we can estimate each block of the Hankel matrix as follows:

$$\begin{aligned} \Phi_D &= \begin{bmatrix} D & CB & CAB & \dots & CA^{T-2}B \end{bmatrix} \\ &= \begin{bmatrix} D & 0 & \dots & 0 \end{bmatrix} + \Phi, \end{aligned} \tag{114}$$

from which we can easily obtain the $D$ matrix and use $\Phi$ to design the controller, as in the rest of the main text. Fortunately, from (113), we see that $\Phi_D$ can also be estimated via (17), i.e.

$$\hat{\Phi}_D := \arg \min_{X \in \mathbb{R}^{d_y \times (T*d_u)}} \sum_{i=1}^{M} \left\| y^{(i)} - XU^{(i)} \right\|_F^2. \tag{115}$$

In particular, we see that even in the case where $D \neq 0$, the estimation error of $D$ does not affect the estimation of $\Phi$ or $\hat{\hat{\Phi}}$. Therefore, the error bound of $N_1, CQ_1, R_1 B$ in Lemma B.2, Lemma B.3, Lemma B.4 still holds. The error of estimating $D$ can be bounded as follows:

**Lemma F.1** (Corollary 3.1 of [51]). *Let $\hat{D}$ denote the first block submatrix in* (115)*, then*

$$\left\| \hat{D} - D \right\| \leq \frac{L}{\sqrt{M}},$$

*where $L$ is a constant depending on $A, B, C, D$, the dimension constants $n, m, d_u, d_y$, and the variance of control and system noise $\sigma_u, \sigma_v, \sigma_w$.*

Therefore, we have proved a bound in the place of Lemma E.1. The rest of the proof will follow the same line as Appendix E.

# G   Bounding $\hat{S}$ when $N_1$ is not diagonalizable

In proving Theorem 5.3, we used the diagonalizability assumption in Lemma D.1. More specifically, it was used in (48) to upper and lower bound $\Lambda^{\frac{m}{2}+j\alpha}(\Lambda^{-\frac{m}{2}-j\alpha})^*$ for $j = 0, 1, \ldots, \frac{p}{\alpha}$, which is reflected in the value of $\bar{\sigma}_s, \underline{\sigma}_s$. In the diagonalizable case, $\Lambda^{\frac{m}{2}+j\alpha}(\Lambda^{-\frac{m}{2}-j\alpha})^*$ is a diagonal matrix with all entries having modulus 1 regardless of the value of $j$. Now in the non-diagonalizable case, we bound $\Lambda^{\frac{m}{2}+j\alpha}(\Lambda^{-\frac{m}{2}-j\alpha})^*$, provide new values for $\bar{\sigma}_s, \underline{\sigma}_s$, and analyze how it affect the final sample complexity.

Consider

$$
\Lambda = \begin{bmatrix}
J_1 & 0 & 0 & \ldots & 0 \\
0 & J_2 & 0 & \ldots & 0 \\
0 & 0 & J_3 & \ldots & 0 \\
\vdots & \vdots & \vdots & \ddots & \vdots \\
0 & \ldots & \ldots & 0 & J_{k_J}
\end{bmatrix}
\tag{116}
$$

where $k_J$ is the number of Jordan blocks, and each $J_i$ is a Jordan block that is either a scalar or a square matrix with eigenvalues on the diagonal, 1's on superdiagona1, and zeros everywhere else.

Without loss of generality, assume $J_1$ is the largest Jordan block with eigenvalue $\lambda$ satisfying $|\lambda| > 1$ and size $\gamma$, then $J_1 = \lambda I + \Gamma$, where $\Gamma$ is the nilpotent super-diagonal matrix of 1's such that $\Gamma^i = 0$ for all $i \geq \gamma$. Therefore, we have

$$
J_1^{\frac{m}{2}+j\alpha} = (\lambda I + \Gamma)^{\frac{m}{2}+j\alpha} = \lambda^{\frac{m}{2}+j\alpha} \sum_{i=0}^{\gamma} \binom{\frac{m}{2}+j\alpha}{i} \frac{\Gamma^i}{\lambda^i},
$$

and

$$
(J_1^{-\frac{m}{2}-j\alpha})^* = \bar{\lambda}^{-\frac{m}{2}-j\alpha}(I + \frac{1}{\bar{\lambda}}\Gamma^*)^{-\frac{m}{2}-j\alpha} = \bar{\lambda}^{-\frac{m}{2}-j\alpha} \left( \sum_{i=0}^{\gamma} \binom{\frac{m}{2}+j\alpha}{i} \frac{(\Gamma^*)^i}{\bar{\lambda}^i} \right)^{-1},
$$

where $\bar{\lambda}$ is the conjugate of $\lambda$. Because $\lambda^{\frac{m}{2}+j\alpha}\bar{\lambda}^{-\frac{m}{2}-j\alpha}$ has modulus 1, we have

$$
\sigma_{\max}(J_1^{\frac{m}{2}+j\alpha}(J_1^{-\frac{m}{2}-j\alpha})^*) \leq \frac{\sigma_{\max}\left(\sum_{i=0}^{\gamma} \binom{\frac{m}{2}+j\alpha}{i} \frac{\Gamma^i}{\lambda^i}\right)}{\sigma_{\min}\left(\sum_{i=0}^{\gamma} \binom{\frac{m}{2}+j\alpha}{i} \frac{\Gamma^i}{\lambda^i}\right)}.
\tag{117}
$$

We can then calculate,

$$
\begin{aligned}
\sigma_{\max}\left(\sum_{i=0}^{\gamma} \binom{\frac{m}{2}+j\alpha}{i} \frac{\Gamma^i}{\lambda^i}\right) &\leq \sum_{i=0}^{\gamma} \binom{\frac{m}{2}+j\alpha}{i} \\
&\leq \gamma \binom{\frac{m}{2}+j\alpha}{\gamma} \\
&\leq \gamma(\frac{m}{2}+j\alpha)^{\gamma} \\
&\leq \gamma(2m)^{\gamma}
\end{aligned}
$$

where the second inequality requires $m \geq 4\gamma$, and the last inequality uses $j\alpha \leq p = m$. To calculate the smallest singular value, note that $\sum_{i=0}^{\gamma} \binom{\frac{m}{2}+j\alpha}{i} \frac{\Gamma^i}{\lambda^i}$ is an upper triangular matrix with diagonal entries being 1, therefore its smallest singular value is lower bounded by 1. Therefore, we have

$$
\sigma_{\max}(J_1^{\frac{m}{2}+j\alpha}(J_1^{-\frac{m}{2}-j\alpha})^*) \leq \gamma(2m)^{\gamma},
\tag{118}
$$

$$
\sigma_{\max}(J_1^{\frac{m}{2}+j\alpha}(J_1^{-\frac{m}{2}-j\alpha})^*) \geq \frac{1}{\gamma(2m)^{\gamma}}.
\tag{119}
$$

As such, the constants $\bar{\sigma}_s, \underline{\sigma}_s$ in (49) need to be modified to

$$
\bar{\sigma}'_s = \bar{\sigma}_s \gamma(2m)^{\gamma},
\tag{120}
$$

$$
\underline{\sigma}'_s = \underline{\sigma}_s \frac{1}{\gamma(2m)^{\gamma}}.
\tag{121}
$$

This will affect the sample complexity calculation step in (99), which will become

$$\epsilon \lesssim \frac{1}{\sqrt{\kappa(\mathcal{G}_{\mathrm{ob}})\kappa(\mathcal{G}_{\mathrm{con}})}L^8 \textcolor{blue}{\gamma(2m)^\gamma}} \min(\sigma_{\min}(\mathcal{G}_{\mathrm{ob}}), \sigma_{\min}(\mathcal{G}_{\mathrm{con}}))(\frac{2|\lambda_k|}{|\lambda_k|+1})^{m/2} \min((|\lambda_k|-1)^2, |\lambda_k|-1)\min(\epsilon_*, 1),$$
(122)

where the only difference is the additional factor in the denominator highlighted in blue. As this factor is only polynomial in $m$, we can merge it with the exponential factor $(\frac{2|\lambda_k|}{|\lambda_k|+1})^{m/2}$ such that

$$\frac{(\frac{2|\lambda_k|}{|\lambda_k|+1})^{m/2}}{\gamma(2m)^\gamma} \geq (\sqrt{\frac{2|\lambda_k|}{|\lambda_k|+1}})^{m/2}\frac{1}{\gamma}(\frac{e\log\frac{2|\lambda_k|}{|\lambda_k|+1}}{8\gamma})^\gamma.$$

Therefore, (100) will be changed into:

$$m \gtrsim \frac{1}{\log\frac{2|\lambda_k|}{|\lambda_k|+1}}\log\frac{\sqrt{\kappa(\mathcal{G}_{\mathrm{ob}})\kappa(\mathcal{G}_{\mathrm{con}})}L^8}{\min(\sigma_{\min}(\mathcal{G}_{\mathrm{ob}}), \sigma_{\min}(\mathcal{G}_{\mathrm{con}}))\frac{1}{\gamma}(\frac{e\log\frac{2|\lambda_k|}{|\lambda_k|+1}}{8\gamma})^\gamma(\log\frac{2|\lambda_k|}{|\lambda_k|+1})^{3/2}\min((|\lambda_k|-1)^2, |\lambda_k|-1)\min(\epsilon_*, 1)},$$
(123)

and lastly, the final simplified complexity on $m$ will be changed into

$$m \gtrsim \max(\alpha, \frac{\log L}{1-|\lambda_{k+1}|}, \frac{\gamma}{|\lambda_k|-1}\log\frac{\kappa(\mathcal{G}_{\mathrm{ob}})\kappa(\mathcal{G}_{\mathrm{con}})L\gamma}{\min(\sigma_{\min}(\mathcal{G}_{\mathrm{ob}}), \sigma_{\min}(\mathcal{G}_{\mathrm{con}}))(|\lambda_k|-1)\epsilon_*}).$$
(124)

which only adds a multiplicative factor in $\gamma$ and the additional $\gamma$ in the log. Given such a change in the bound for $m$, the bound for other algorithm parameters $p, q, M$ is the same (except for the changes caused by their dependance on $m$).

# H  Additional Helper Lemmas

**Lemma H.1.** *Given Assumption 5.1 is satisfied, then $N_1, CQ_1$ is observable, $N_1, R_1B$ is controllable.*

*Proof.* Let $w$ denote any unit eigenvector of $N_1$ with eigenvalue $\lambda$, then

$$N_1w = \lambda w \quad \Rightarrow \quad AQ_1w = Q_1N_1w = \lambda(Q_1w).$$

Therefore, $Q_1w$ is an eigenvector of $A$. As $A, C$ is observable, by PBH test, this leads to

$$\|CQ_1w\| > 0.$$

By PBH Test, this directly leads to $(N_1, CQ_1)$ is observable. The controllability part is similar.

$\square$

**Lemma H.2** (Gelfand's formula). *For any square matrix $X$, we have*

$$\rho(X) = \lim_{t\to\infty}\|X^t\|^{1/t}.$$

*In other words, for any $\epsilon > 0$, there exists a constant $\zeta_\epsilon(X)$ such that*

$$\sigma_{\max}(X^t) = \|X\| \leq \zeta_\epsilon(X)(\rho(X) + \epsilon)^t.$$

*Further, if $X$ is invertible, let $\lambda_{\min}(X)$ denote the eigenvalue of $X$ with minimum modulus, then*

$$\sigma_{\min}(X^t) \geq \frac{1}{\zeta_\epsilon(X^{-1})}\left(\frac{|\lambda_{\min}(X)|}{1+\epsilon|\lambda_{\min}(X)|}\right)^t.$$

The proof can be found in existing literatures (e.g. [18].

**Lemma H.3.** *For two matrices $H, E$, $\frac{1}{2}H^*H - E^*E \preceq (H+E)^*(H+E) \preceq 2H^*H + 2E^*E$.*

*Proof.* For the upper bound, notice that

$$2H^*H + 2E^*E - (H+E)^*(H+E) = H^*H + E^*E - H^*E - E^*H = (H-E)^*(H-E)^* \succeq 0$$

Therefore, we have

$$(H+E)^*(H+E) \preceq 2H^*H + 2E^*E. \tag{125}$$

For the lower bound, we have

$$H^*H = (H+E-E)^*(H+E-E)^* \preceq 2(H+E)^*(H+E) + 2E^*E$$

where in the last inequality, we used (125). Rearrange the terms, we get the lower bound.

$$\frac{1}{2}H^*H - E^*E \preceq (H+E)^*(H+E).$$

$\square$

**Theorem H.4** (Rouche's theorem). *Let $D \subset \mathbb{C}$ be a simply connected domain, $f$ and $g$ two meromorphic functions on $D$ with a finite set of zeros and poles $F$. Let $\gamma$ be a positively oriented simple closed curve which avoids $F$ and bounds a compact set $K$. If $|f - g| < |g|$ along $\gamma$, then*

$$N_f - P_f = N_g - P_g$$

*where $N_f$ (resp. $P_f$) denotes the number of zeros (resp. poles) of $f$ within $K$, counted with multiplicity (similarly for $g$).*

For proof, see e.g. [2].

# I  Indexing

For the convenience of readers, we provide a table summarizing all constants appearing in the bounds.

Table 1: Lists of parameters and constants appearing in the bound.

| Constant | Appearance | Explanation |
|---|---|---|
| $T$ | Section 1 | Length of each roll out. |
| $M$ | Section 3 | Number of roll outs. |
| $m$ | Section 3 | number of open-loop steps for converging to unstable state space. |
| $p, q$ | Section 3 | dimension of Hankel matrix to be estimated. |

Table 2: System theoretic parameters.

| Constant | Appearance | Explanation |
|---|---|---|
| $H$ | (5) | Ground truth Hankel matrix |
| $\mathcal{O}, \mathcal{C}$ | (6) | factorization of $H$. |
| $\tilde{\mathcal{O}}, \tilde{\mathcal{C}}$ | (7) | rank-$k$ approximation of $\mathcal{O}, \mathcal{C}$, respectively. |
| $\tilde{H}$ | (8) | rank-$k$ approximation of $H$. |
| $\hat{H}$ | (18) | numerical approximation of $H$. |
| $\hat{\tilde{H}}$ | (19) | rank-$k$ approximation of $\hat{H}$. |
| $\hat{\mathcal{O}}, \hat{\mathcal{C}}$ | (20) | factorization of $\hat{\tilde{H}}$. |
| $\mathcal{G}_{\mathrm{ob}}$ | Section 5 | Observability Gramian |
| $\mathcal{G}_{\mathrm{con}}$ | Section 5 | Controllability Gramian |
| $\tilde{o}_{\min}(m)$ | Lemma D.1 | a positive real number such that $\sigma_{\min}(\tilde{\mathcal{O}}) \geq \tilde{o}_{\min}(m)$. |
| $\tilde{c}_{\min}(m)$ | Lemma D.1 | a positive real number such that $\sigma_{\min}(\tilde{C}) \geq \tilde{c}_{\min}(m)$. |
| $\alpha$ | Appendix C | Controllability index. |
| $C_C, C_B$ | (48) | modified controllability and observability matrices. |
| $\zeta_\epsilon(\cdot)$ | Lemma H.2 | Gelfand constant for the norm of matrix exponents |

Table 3: Shorthand notations (introduced in proofs).

| Constant | Appearance | Explanation |
|---|---|---|
| $\underline{\sigma}_s$ | Lemma D.1 | $\underline{\sigma}_s := \frac{\sigma_{\min}(\mathcal{G}_{\mathrm{ob}})}{\sigma_{\max}(\mathcal{G}_{\mathrm{con}})} \sigma_{\min}(P_1)^2$ |
| $\overline{\sigma}_s$ | Lemma D.1 | $\overline{\sigma}_s := \frac{\sigma_{\max}(\mathcal{G}_{\mathrm{ob}})}{\sigma_{\min}(\mathcal{G}_{\mathrm{con}})} \sigma_{\max}(P_1)^2$ |
| $\epsilon_N$ | Lemma B.2 | $\epsilon_N := \frac{4Lc_2\epsilon}{c_1 \tilde{o}_{\min}(m)}$ |
| $\epsilon_C$ | Lemma B.3 | $\epsilon_C := 2\left( \frac{4c_2 L}{c_1 \tilde{o}_{\min}(m)} + \frac{1}{\tilde{c}_{\min}(m)} \right)\epsilon$ |
| $\epsilon_B$ | Lemma B.4 | $\epsilon_B := 8\left( \frac{c_2 L}{c_1 \tilde{c}_{\min}(m)} + \frac{1}{\tilde{o}_{\min}(m)} \right)\epsilon$ |
| $\epsilon_*$ | Assumption 5.2 | |
| $\epsilon$ | Lemma B.1 | $\epsilon := 2\hat{\epsilon} + 2\tilde{\epsilon}$ |
| $\hat{\epsilon}$ | Theorem C.2 | $\hat{\epsilon} := \frac{8\sigma_v \sqrt{T(T+1)(d_u+d_y)\min\{p,q\}\log(27T/\delta)}}{\sigma_u \sqrt{M}}$ |
| $\tilde{\epsilon}$ | Lemma C.3 | $\tilde{\epsilon} := L^3 (\frac{|\lambda_{k+1}|+1}{2})^m$ |

