# OpenReview forum: "Stabilizing LTI Systems under Partial Observability: Sample Complexity and Fundamental Limits"
_NeurIPS.cc/2025/Conference — NeurIPS 2025 poster_

### Official Review · Reviewer_HW5z · 2025-06-21

**Clarity:** 3
**Significance:** 3
**Originality:** 3
**Rating:** 5
**Confidence:** 3

**Summary:**

The paper proposes an approach to designing a stabilizing controller for partially observable linear time-invariant systems with disturbances. Based on a number of (partially) observed trajectories, obtained by applying random Gaussian control inputs, the unstable system components are approximated for a given rank. By raising the system matrix to a given power, the influence of the unstable eigenvalues is amplified, which is crucial for their identification. This is referred to in the paper as lifting the Hankel matrix. The resulting approximation is then used to design the stabilizing controller. The paper provides probabilistic guarantees that the derived controller actually stabilizes the system, and clearly identifies the influence of the system parameters as well as the hyperparameters of the algorithm.

**Questions:**

In practice, an unstable system will be only be observed or simulated over a short time horizon. This situation seems to merit some discussion. In my understanding, it suffices, roughly speaking, that m is as large as the unstable dimension k and T is larger than 3m. Then any probability can be guaranteed by making M large enough.

The claim about boundedness in line 67 could be clarified: In a strictly finite observation scenario (which is considered here), Hankel matrices are finite and bounded.

**Ethical Concerns:**

["NO or VERY MINOR ethics concerns only"]

**Final Justification:**

The author response answered my questions. I maintain my judgement that this is a paper that should be accepted.

**Limitations:**

I did not see any discussion of numerical difficulties that might arise from observations of unstable systems, the inversion of the matrices used in the algorithm, etc. In fairness to the authors, I must admit that the paper is already heavily packed and I have no suggestions on how to make more room for such an analysis.

Note that I did not check the lengthy math appendix in detail, but I have no reason to doubt its correctness.

**Quality:**

3

**Strengths And Weaknesses:**

Strengths:
- The paper extends existing work from fully observable to partially observable systems.
- The probabilistic guarantees are clear and expose the influence of critical parameters, both of the system and the algorithm.
- The sample complexity is relatively low. As in prior related work, the system dimension does not directly influence it. In particular, even modest increases in the number of trajectories substantially raise the probability of success.

Weaknesses:
- The paper is somewhat technical and compact. I admit this is somewhat in the nature of the contribution and not easy to improve. For example, it took me a while to remember what m,M,p,q stand for. Since the result at the end proposes p=q=m, this could be mentioned earlier to ease some of the cognitive load. Reading Algorithm 1 requires referring to 8 equations that are listed on the next page. An intuitive explanation would be just as useful.
- The paper is largely inspired by [50], where, thanks to full observability, only a single trajectory is required.

---

> ### Author Rebuttal · Authors · 2025-07-30
>
> We thank the reviewer for the careful and thorough review and provide the explanation below:
>
> - Trajectory length: The reviewer is absolutely correct that by picking a minimally satisfactory $T$ with $m \approx k$ and $T \approx 3m$, simulations have shown we can increase the probability of getting accurate system identification and stabilization policy almost surely by increasing $M$. This roughly matches the order in the theorem ($m$ scales linearly with $k$), though we do acknowledge that the theorem has very large numerical constants and $m$ needs to depend on other constants as well. We have added discussions on this in the paper.
>
> - Bounded Hankel around line 67: The reviewer is right that we should have been more careful with word choice. While it is true that in a strictly finite observation scenario, Hankel matrix is always finite and bounded, estimating Hankel matrices accrately becomes much more difficult when the system is unstable. As the $(i,j)$-th entry of the Hankel matrix is $CA^{m+i+j-2} B$ for some $i,j \in \mathbb{N}$, if $A$ is stable, $\Vert CA^{m+i+j-2} B\Vert$ decreases with increase in $i,j$, so the error of each entry decreases as the trajectory gets longer. However, if $A$ is unstable, $\Vert CA^{m+i+j-2} B\Vert$ increases, so any error in system estimation of $A$ (and by extension $B,C$) would be amplified, making the problem more difficult. We will clarify this in the revision.
>
> We hope the above have answered your questions and provided some more context. Please let us know if you have further questions or suggestions.

---

> > ### Comment · Reviewer_HW5z · 2025-08-07
> >
> > Thank you for your detailed response. Your comments clarify the issues I raised and I maintain my positive evaluation.

---

### Official Review · Reviewer_EFNX · 2025-06-27

**Clarity:** 4
**Significance:** 3
**Originality:** 4
**Rating:** 5
**Confidence:** 4

**Summary:**

This paper studies the problem of stabilizing unknown *partially observable* LTI systems, a natural and necessary extension of previous results in the *fully observable* setting. The authors propose **LTS-P**, a novel algorithm that efficiently stabilizes the system via isolation of  the unstable system modes using compressed SVD on a "lifted" Hankel matrix, and via a robust controller design under a small-gain-type assumption on the stable residue. The algorithm features a reduced sample complexity that only depends on the number of unstable modes and is thus independent of the system dimension. Preliminary simulation results confirm its outperformance over existing algorithms.

**Questions:**

1. How do system parameters ($\epsilon_*$, $L$) scale with the system dimension $n$? See above for details.
2. Additional experimental results would be appreciated.

**Ethical Concerns:**

["NO or VERY MINOR ethics concerns only"]

**Quality:**

3

**Strengths And Weaknesses:**

Strengths:
1. The paper is overall well-written and well-organized. It provides an abundance of intuition via simplified settings and comparisons against known results in the main text, while retaining mathematical rigor in the appendix by providing additional technical details.
2. The research question is of great theoretical interest, as it's a natural follow-up of several recent works in the fully observable setting.
3. The algorithm design idea builds upon a solid ground that isolates the unstable modes of the system, but also provides new insights that leverages the transfer function, "lifted" Hankel matrix and small-gain-type arguments to properly handle partially observable systems.
4. The theoretical results are significant, and the proofs are checked to be sound.

Weaknesses:
1. Assumption 5.2, though coming with a detailed comment on its necessity, still seems a little suspicious. Specifically, it is unclear how $\epsilon_*$ scales with the system dimension $n$. The order of sample complexity could be made more convincing if the growth rate of $\log \frac{1}{\epsilon_*}$ (and also $L$) is, at least, illustrated with a representative family of systems with increasing dimensions.
2. The simulation results presented in Appendix A is, at best, preliminary and illustrative. Since $MT$ is selected as the complexity metric, it is natural to ask if there are any potential tradeoffs between $M$ and $T$, which is not reflected in the simulations. Instead, performance is only compared in a restrictive setting, with either $M$ or $T$ fixed. The experimental results could be more convincing through more settings and more results.
3. Some notations, though easy to judge from context, are not defined before first use. For example, $\hat{\Phi}_i$ in eq. (18) seems to be the $i$th block of $\hat{\Phi}$.
4. A few typos are observed, including "ovservable" -> "observable" (line 32), "folloing" -> "following" (line 175), etc.

---

> ### Author Rebuttal · Authors · 2025-07-30
>
> We thank the reviewer for the careful and thorough review and provide the explanation below:
>
> - Scale of constants: The constant $\epsilon_*$ is a stability margin linking the necessary condition for the existence of a stabilizing controller, i.e. $\Vert K(z)F_K(z)\Vert_{H\_{\infty}} < \frac{1}{\Vert \Delta(z)\Vert_{H\_{\infty}}}$ as in Lemma 3.1 and the actual bound on $\Vert K(z)F_K(z)\Vert_{H\_{\infty}}$ in Assumption 5.2 which requires some margin on top of Lemma 3.1. If we interpret $\epsilon_*$ to be in the same order as $\Vert\Delta(z)\Vert_{H_{\infty}}$, then it does not inherently scale with $n$ as $\Vert \Delta(z)\Vert_{H_{\infty}}$ characterizes the input-output frequency response, related to the input-output dimension, not directly related to the internal state dimension.
>     On the other hand, $L$ covers all other constants needed in Theorem 5.3, it upper bounds the matrix norm of all of the following matrices $\Vert A\Vert, \Vert B\Vert,\Vert C\Vert, \Vert N_1\Vert, \Vert R_1 B\Vert, \Vert CQ_1\Vert$. If we take $A$ to be from a large-scale networked LTI systems, i.e. with very large $n$ but $A$ admits a sparsity pattern to a sparse graph, then $\Vert A\Vert$ scales with the maximum degree of the graph (assuming each entry of $A$ is $O(1)$).
>
>     The dependence on $n$ is also shown in simulation, i.e. as we increased the state dimension $n$ with randomly generated matrices, the length of trajectories $T$ and number of trajectories $M$ stayed the same.
>
>     Finally, we point out that even if $\epsilon_*, L$ depends on $n$ polynomially, its effect on the final complexity bound is only $\log n$ as $\epsilon_*,L$ only shows up in $\log$ terms.
>
> - $M,T$ tradeoff: Thanks for your suggestions. The $M$ and $T$ trade off is indeed the idea we want to analyze in the simulation. Although the preliminary result shows that fewer trajectories (smaller $M$) requires longer trajectories (larger $T$) and shorter trajectories (smaller $T$) requires more trajectories (larger $M$), we did only include one simulation for a single fixed $T$ and varying $M$ and another simulation for a single fixed $M$ and varying $T$. We did not include more results because it seems the trend of the plots look similar to what is present. However, the reviewer is right in pointing out that there might be value in collecting the percentage of successful stabilization for a fixed $n$ across different $M,T$ pairs. We are currently completing this simulation and will include that in the final version.
>
> We hope the above have answered your questions and provided some more context. Please let us know if you have further questions or suggestions.

---

> > ### Comment · Reviewer_EFNX · 2025-08-01
> >
> > Thank you for your response, and also for this great work! I'll keep my positive evaluation of this paper.

---

### Official Review · Reviewer_H1gh · 2025-07-02

**Clarity:** 2
**Significance:** 3
**Originality:** 3
**Rating:** 4
**Confidence:** 3

**Summary:**

This paper gives an algorithm for learning stabilizing controllers for an unstable LTI systems. The key insight is that the unstable dynamics can be learned via a specialized subspace identification algorithm. Then, as long as the stable part has a sufficiently small $H_{\infty}$ norm,  a robust controller designed for the unstable component will stabilize the system with high probability.

**Questions:**

- What happens if there are marginally stable eigenvalues?
- Can you do the simulations with the theoretical choices for $m$, $p$, and $q$?
- Do you run into numerical problems when $m$ is large?
- Can Assumption 5.2 be removed completely?

**Ethical Concerns:**

["NO or VERY MINOR ethics concerns only"]

**Final Justification:**

The authors did a good job convincing me that the contribution was reasonably strong.

However, there are still issues remaining that were not adequately addressed:
- Assumption 5.2 does not appear to be necessary. Only necessary when the stable part is not identified.
- I still think the presentation of the parameter choices in the theorem is not good.
- The work does not not adequately address issues of non-zero initial conditions and process noise.

Based on the limitations above, I don't really want to raise beyond a weak accept.

**Limitations:**

Yes.

**Paper Formatting Concerns:**

Nothing major, but some equations go into the margins.

**Quality:**

3

**Strengths And Weaknesses:**

# Strengths
- This gives at least a partial solution to a long-standing problem in learning-based control: learning a stabilizing controller for partially observed systems.
- The writing is reasonably clear.

# Weaknesses
- Theorem 5.3 appears to be stated incorrectly. Namely, I'm guessing (based on Lemma B.1 and Theorem 3.2) it should be that for all $\delta\in (0,1)$ the system is stabilized with probability at least $1-\delta$. The theorem is currently stated as "for some $\delta$" the system is stabilized with probability at least $\delta$.
- The work doesn't give a full solution to the stabilization problem: It really only works for sufficiently small stable parts. (I.e. Assumption 5.2 is required.) While an argument about how to extend the method is sketched, a complete solution that works for all stabilizable and detectable LTI systems would be desirable.
- The paper is somewhat inconsistent in the treatment of process noise. Namely, process noise is included in (14) and (26), but is  not used in the (1) or the main theory.  My guess is that the problem would be substantially harder with process noise.  Namely, (31) would include terms that get amplified through the unstable dynamics, leading to substantially larger errors in the Hankel matrix estimation.
- Similarly, the paper assumes that the state can be reset to 0 at the beginning of each trajectory. If this were not the case, similar to  the problem with process noise, unobserved, non-zero initial conditions would make the Hankel matrix errors much larger.
- The choices of $m$, $p$, and $q$ used in the simulations are inconsistent with the theory.
- The choice of $m$ in Theorem 5.3 requires knowledge of the eigenvalues, and properties of various Gramians. As written, this does not appear to be very useful, since it requires choosing a parameter of the algorithm, $m$, based on properties of  the unknown system. The result could be stated a bit differently, so that for a given $m$, the algorithm can stabilize the system provided the bounds on the eigenvalues and Gramians holds. I.e. for a given $m$, the algorithm would work for all systems of a particular class.

---

> ### Author Rebuttal · Authors · 2025-07-30
>
> We thank the reviewer for the careful and thorough review and provide the explanation below:
>
> - Regarding $\delta$: We thank the reviewer for pointing out the typo in the statement of Theorem 5.3. It should have been "with probability at least $1-\delta$", instead of "with probability at least $\delta$".
>
> - Assumption 5.2 and no marginally stable eigenvalue: Both assumptions are necessary for this unstable+stable decomposition to work. As discussed at the end of Section 5, Assumption 5.2 comes from the Small Gain Theorem, which is an if and only if condition. If Assumption 5.2 is not true, it can be shown that no matter what control law $K(z)$ the user picks, there always exists some $\Delta(z)$ that would make the system unstable [53]. Regarding marginal stable eigenvalues, as our results depend on the separation of the unstable and stable components, we do not allow marginal eigenvalues. That being said, we do allow the eigenvalue to be arbitrarily close to $1$, and its impact on sample complexity is characterized in our sample complexity bounds, discussed under the ''Dependence on system theoretic parameters'' paragraph after Thm 5.3.
>
>     Regarding your point ''The work doesn't give a full solution to the stabilization problem:'', we wish to point out that the learn-to-stabilize problem has been shown as a challenging problem with various exponential lower bounds proven in [9] [48] and further in paper ''Learning to Control Linear Systems can be Hard''. Our result essentially carved out a significant subset of problems (those satisfying assumptions in the paper) for which learning-to-stabilze can be easy. In fact, our assumptions is minimal in order to break the hardness of stabilization proposed in [9,48] if one only stabilizes the unstable subspace of the system. While we do acknowledge there is more to do towards ``full solution to the stabilization problem'', our results already make a significant step towards this problem as all prior results actually show learn-to-stabilize is hard.
>
> - Process noise: The paper's theoretical guarantee works when there is no process noise, but observation noise is allowed (Equation (1), (14), (26) all include the observation noise $v_t \sim \mathcal{N}(0,\sigma_v)$).  Process noise is only included in Equation (26) as $w_t$ to show that the proposed method works under process noise in simulation. As the reviewer points out, introducing process noise would enlarge the error of estimating $\mathcal{H}$ with an additional error term in Equation (31).
>
>     We clarify that the main technical novelty of the paper is the stable + unstable decomposition (including techniques to separate the unstable component from the lifted Hankel), whereas the system ID part to get the Hankel is not the main focus. We chose to assume process noise is $0$ to keep the system ID part cleaner, avoiding complicating the paper with unnecessary details.
>     If process noise are to be included, then as long as the Hankel estimation error can be bounded, the rest of the proof would work in a similar way. Here is the outline of what the proof would look like:
>
>     If there is process noise $w_t$, a matrix $\Psi = [0, A, A^2, \dots,A^T]$ will have to appear in the proof, so the estimation error of $\hat{\tilde{H}}$ will contain a term that increases exponentially with $T$.
>     This is to be expected, as process noise would be amplified by the unstable part of $A$. Once we bound the Hankel estimation, the rest of the proof will be exactly the same as in this paper. In order to finish proof in this direction, a fine balance of $T$ needs to be selected, so that it needs to minimize the current terms in (109), but also not make a term containing $\Psi$ blow-up, so more weight needs to be applied to the number of trajectories $M$, instead of $T$. However, this fine balance seems only to be present on the above theoretical analysis, as in simulation, we do include process noise we see that (a) the number of samples needed for stabilization does not increase with $n$, confirming our approach works under process noise; (b) a longer trajectory still helps with stabilization even with process noise, meaning the fine-balance regarding the choice of $T$ is not present in our simulation. We leave how to improve the theoretical analysis to incorporate process noise as a future direction of this paper.
>
> - Initial condition: The paper does assume the system is reset to $0$ at the start of each trajectory, but this is merely for the simplicity of proof. For simplicity, we will set $D = 0$ again in this part of the discussion. If $x_0 \neq 0$, then $y_{t+1} = CA^t x_0 + \sum_{i=0}^{t}CA^{t-i}B u_i$, with the additional term $CA^t x_0$ known as the free-response term. We can pick some $p$ such that $k < p <T$ and define $U_p = [u_0,\dots, u_{T-p}], U_f = [u_p, \dots, U_T]$ and $Y_p = [y_0, \dots, y_{T-p}], Y_f = [y_p \dots, y_T]$. We can then project the future outputs orthogonally to the row space of $Y_p$ by $\Pi^{\perp} = I- Y_p^\top (Y_p Y_p^{\top})Y_p$. Then, we can use the observation $\tilde{Y}_f = Y_f \Pi^\perp$ for the proposed algorithm. This is a quite popular method used in conventional control methods, such as N4SID. For details on how and why it works, please refer to Section 3 of ''N4SID: Subspace Algorithms for the Identification of Combined Deterministic-Stochastic System'' by Peter Van Overschee and Bart De Moor. This is indeed an important issue; we have added more discussion on it in the paper.
>
> - Inconsistencies in $m,p,q$ btw theory and simulation: Theorem 5.3 offers the order of $m,p,q$, not the exact value we should pick (the $\asymp$ means only ignoring numeric constant as defined at the start of Section 2 ).
>     These expressions provide theoretical insights, showing how $m,p,q$ should depend on system size $n$, unstable subspace $k$, observability, and contrability of the system. Making an exact computation of the omitted numerical constants might offer conservative values, as throughout the proof, many terms are relaxed and merged with the leading order term, making the final order-wise dependence clean at the cost of a conservative numerical constant. For this reason, in experiments, we do not follow the exact theoretical values, but follow trial and error instead. This is common practice in many theoretical literature in control and optimization, where the parameters chosen in the Theorems are conservative, and simulations use much more relaxed parameters.
>
> - Usefulness of bounds for $m$: We disagree that ``the choice of $m$ ... does not appear to be very useful''. The choice of $m$ eventually translates to the sample complexity requirements $ (3m+2)M$, which answers fundamental questions of how the sample complexity of stabilization depends on system theoretic parameters like $n,k$, eigenvalue gap, controllability gramian etc. These bounds provide insights like some systems are inherently easier to stabilize than others. For example, we see it is harder to separate the stable system from the unstable system if there are marginally stable eigenvalues. This is why $m$ is proportional to $\frac{1}{1-|\lambda_{k+1}|}$ and $\frac{1}{|\lambda_{k}|-1}$. Moreover, the less observable or controllable a system is, the smaller $\sigma_{\min}(\mathcal{G}\_{\text{ob}})$ or $\sigma_{\min}(\mathcal{G}\_{\text{con}})$ is, respectively, and the larger $\kappa(\mathcal{G}\_{\text{ob}})$ or $\kappa(\mathcal{G}\_{\text{con}})$ is, respectively. Therefore, the less observable or controllable a system is, the harder it is to stabilize the system.
>
>     We acknowledge that in practice, the theoretical values for $m$ cannot be exactly computed, as it depends on the parameter of the unknown system. However, we point out that as long as we plug in conservative estimates of those parameters into $m$ (e.g. lower bounds for $\sigma_{\min}(\mathcal{G}\_{\mathrm{ob}}),\sigma_{\min}(\mathcal{G}\_{\mathrm{con}})$), the theorem still works. Further, as in many theoretical papers, the selection of those parameters in experiments may differ from those in theory.
>      In fact, in our experiments, the choice of $m$ is very forgiving. Unless $m$ is too small (so we did not converge to unstable subspace enough) or too close to $T$ (so we did not leave enough samples for system estimation), we generally get quite good system estimation. Throughout the simulation, we did not need to do much hand-picking on $m,p,q$. We have added more discussions on this issue in the paper.
>
> We hope the above have answered your questions and provided some more context. Please let us know if you have further questions or suggestions.

---

> > ### Comment · Reviewer_H1gh · 2025-08-05
> >
> > The rebuttal has convinced me that the paper is solving an important sub-class of the general learning to stabilize problem. Based on this, I will raise my score to a weak accept.
> >
> >  But I would like to push back and clarify a few points:
> > - Assumption 5.2 is **only** necessary in the case that just the unstable component of the transfer function is being learned. This is in order to apply the small gain theorem for the stable component. But if the stable component were also learned, Assumption 5.2 should not actually be necessary.
> > - This is what I meant by not having Assumption 5.2. The writing in the paper is fairly careful to note that the necessity arises in the specific case that only the unstable part is used.
> > - While I agree that knowing the requirements on $m$ is useful, the authors seemed to have somewhat missed my point on my comment about it. Of course, understanding what the $m$ should satisfy is valuable. My point was about how it is written. I was encouraging the authors to rewrite the theorem in a way that didn't require making a choice of an algorithm parameter based on unknown quantities.
> >     * My original suggestion was to reframe the statement so that for a particular choice of $m$, the algorithm guarantees stability for all systems in a particular class. (Found based on the calculations for $m$)
> >     * The suggestion from the rebuttal would be to approximate/bound the parameters used to calculate $m$. This seems like more work, if you actually wanted to do it rigorously. But it would be perfectly valid, if it were done.
> > - I'm not convinced that the argument from Van Overschee and De Moor actually covers the non-zero initial condition problem. Namely, as discussed in the referenced paper, the signals are required to be quasi-stationary, which fails in the unstable case.
> >
> > My issues with the mismatch between the theory and the parameters used in numerics is more an issue of preference. I recognize that the practices in the paper are common. But it does point to gaps between what is observed in practice and what can be proved theoretically, and I do think it is valuable to be up front about those gaps.

---

> > > ### Author Response · Authors · 2025-08-06
> > >
> > > We thank the reviewer for the timely response and suggestions and provide some clarifications below:
> > > - The authors agree that Assumption 5.2 is **only** necessary if we only learn the unstable component of the transfer function. As pointed out by the reviewer, the submission is ''fairly careful to note that the necessity arises in the specific case that only the unstable part is used''. In the revised paper, we will make it even clearer.
> > > - The authors thank the reviewer for the suggestion on the statement of Theorem 5.3. We are working on formulating a corollary, which states that given a particular choice of $m$, what are the conditions on the system parameters (e.g. the unstable dimension $k$, control input dimension $d_u$, output dimension $d_y$, controllability/observability grammain parameters etc), so that algorithm can assure stabilization of the system with high probability. We will add this corollary to the revised version.
> > > - The authors acknowledge that the theoretical guarantee for non-zero initial position in Van Overschee and De Moor currently only applies to a quasi-stationary system, and the estimation of the Hankel matrix will contain a new error term that is again exponential in $T$, as in the case of process noise, precisely for the reason the reviewer pointed out. The theoretical analysis of non-zero initial position is indeed non-trivial, and we leave this as a future direction of this work. We will acknowledge this point (along with points about process noise) in the revised manuscript.

---

### Official Review · Reviewer_sNGj · 2025-07-03

**Clarity:** 4
**Significance:** 3
**Originality:** 3
**Rating:** 5
**Confidence:** 3

**Summary:**

This paper presents a novel method for obtaining a stabilizing controller for a partially observed discrete-time LTI system, and analyzes the sample complexity of the resulting algorithm. The method is based on estimating the parameters of a lifted Hankel matrix for the system using multiple trajectories with iid control inputs. The estimated Hankel matrix allows for estimating the open-loop transfer function for only the unstable part of the system, which is then used to construct a stabilizing controller for the entire system. In the case where the number of unstable eigenvalues $k$ is much smaller than the state dimension $n$, this decomposition into unstable and stable parts yields better sample complexity than applying previous results on system identification to this problem. Experiments on randomly generated systems validate that the proposed method learns a stabilizing controller with less data in practice.

**Questions:**

- In the case when $k$ is unknown, the paper describes a procedure for finding $k$ based on the singular values of the estimated Hankel matrix which assumes that the lifting parameters $p$, $q$, and $m$ are sufficiently large with respect to $k$. What happens when this is not the case? Is it possible to mistakenly believe that the true $k$ has been identified (i.e. find a large spectral gap in the singular values that does not correspond to $k$)? I would be inclined to raise my score if I have misunderstood this limitation.

- Do the simulations in appendix A use constant $B$, $C$, and $D$ matrices while randomizing $A$?

- If the simulation results instead plotted the fraction of randomized systems that are successfully stabilized for a given amount of data, can we recover $\delta$ in theorem 5.3?

**Ethical Concerns:**

["NO or VERY MINOR ethics concerns only"]

**Final Justification:**

The authors did a nice job of rebutting my questions/concerns as well as those of the other reviewers. This is a strong paper and I recommend that it should be accepted.

**Limitations:**

Yes

**Paper Formatting Concerns:**

No issues.

**Quality:**

4

**Strengths And Weaknesses:**

Strengths: This is a strong paper with a good motivation and clear development of the technical ideas underlying the proposed method. It tackles a problem of practical interest that has not been targeted in prior works to the best of my knowledge. It is also very well-organized and relatively easy to follow. I especially appreciated the notation index that not only defines each term but cites the equation / lemma where each appears! Providing a code repository will make it easy for other researchers to use the method in their own applications.

Weaknesses: One important limitation of the paper is that the method assumes $k$ is known. A procedure for estimating it is presented in the paper, but this method only works when the lifting parameters $p$, $q$, and $m$ are sufficiently large with respect to $k$, which is problematic when $k$ is truly unknown.

Minor weaknesses:
- Above equation (24), I believe “$\bar{C}$ and $\bar{B}$” should read “$\hat{C}$ and $\hat{B}$”
- It would be helpful if the captions in figure 1 and figure 2 were more detailed, including phrases like “at different noise levels $\sigma$” and “with rollouts of length $T=4n$” (figure 1) or “with rollouts of length $T=70$ (figure 2)
- Line 498: trails -> trials

---

> ### Author Rebuttal · Authors · 2025-07-30
>
> We thank the reviewer for the careful and thorough review and provide the explanation below:
>
> - $\bar{C}$ and $\bar{B}$: The notation $\bar{C}$ and $\bar{B}$ above Equation (24) are defined in line 202 and 203, which are our estimates for $C,B$ if the Hankel were exactly known. As the Hankel itself is estimated in the actual Algorithm, Eq. (24) are estimated versions of $\bar{C},\bar{B}$.
>
> - Unknown $k$: What we can prove is that (a) the difference between the lifted Hankel $H$, and $\tilde{H}$ is exponentially small (Lemma C.3); (b) $\tilde{H}$ is formed by a product of a $d_y p\times k$ and a $k\times d_u q $ matrix, so its rank is at most $\min(d_y p, d_u q,k)$; (c) All non-zero singular values of $\tilde{H}$ is exponentially large in $m$. Therefore, when $p,q,m$ are large, from (b) (c) we know $\tilde{H}$ will have $k$ exponentially large (in $m$) singular values and other singular values are zero. Then from (a) we know $H$ will have $k$ exponentially large singular values, and all other singular values are exponentially small. This is the singularvalue gap our paper was referring to and how $k$ can be identified from this gap. For your question where $m,p,q$ are not large enough, there are two cases. Case I: If $p,q$ are not large enough such that the rank of $\tilde{H}$ is smaller than $k$ (i.e. $d_yp<k$ or $d_u q<k$), then by (a) (c) all singular values of $H$ will be exponentially large, and the singular value gap will not be observed. Case II: In the second case, where $m$ is not large enough, then the distinction between the ''exponentially large in $m$'' terms and ``exponentially small in $m$'' will not be obvious, so a large gap is not obvious. To conclude: when $p,q,m$ are not large enough, the singular value gap will not be obvious so one won't mistakenly believe the true $k$ would be identified. Actually, the above discussion also sheds light on how to tune $p,q,m$ to identify the gap.
>
> - Simulation setup: We thank the reviewer for pointing out the issue with the setup in the simulation. In the simulation, $B$ and $C$ are both initialized randomly, where each entry is sampled from an independent uniform distribution between 0 and 1. we still assumed $D = 0$ as discussed in Section 2. Although we discussed how to tackle the case when $D \neq 0$ in Appendix $F$, We did not have non-zero $D$ in the simulation.
>
> - Regarding $\delta$: Yes, if we run the simulation a sufficient number of times for fixed $m,p,q$, we can recover the value of $\delta$. In Theorem 5.3 and Theorem C.2, we use $\delta$ to show that longer trajectories would have better chances of finding accurate Hankel estimation.
>
> We hope the above have answered your questions and provided some more context. Please let us know if you have further questions or suggestions.

---

> > ### Comment · Reviewer_sNGj · 2025-08-06
> >
> > Thank you for the nice rebuttal, including addressing my main concern (unknown $k$).  Since I already voted to accept and it seems like I'm in line with the other reviewers I'm not inclined to change my score and I don't have any additional comments/questions.

---

### Decision · Program_Chairs · 2025-09-17

**Decision:**

Accept (poster)

**Comment:**

The authors give a provable algorithm for stabilizing an unknown partially observable linear time-invariant system, scaling only with respect to the dimension of the unstable subpsace.

Reviewers agreed that this is a technically sound and novel contribution to an important problem, and after discussion, all were in favor of acceptance. As pointed out by H1gh, the result could potentially be technically improved in some ways (more general setting with initial conditions and process noise); though the paper is still significant in carving out a first family of problems where "learn-to-stabilize" is provably efficient.

I remind the authors to incorporate the suggestions and clarifications into the final paper.